# Loss of aquaporin-4 results in glymphatic system dysfunction via brain-wide interstitial fluid stagnation

**Ryszard Stefan Gomolka[1], Lauren M Hablitz[2], Humberto Mestre[2,3], Michael Giannetto[2], Ting Du[2,4], Natalie Linea Hauglund[1], Lulu Xie[2], Weiguo Peng[1,2], Paula Melero Martinez[1], Maiken Nedergaard[1,2]\*, Yuki Mori[1]\***

[1]Center for Translational Neuromedicine, University of Copenhagen, Copenhagen, Denmark; [2]Center for Translational Neuromedicine, University of Rochester Medical Center, Rochester, United States; [3]Department of Neurology, University of Pennsylvania, Philadelphia, United States; [4]School of Pharmacy, China Medical University, Shenyang, China

**\*For correspondence:**
nedergaard@sund.ku.dk (MN);
yuki.mori@sund.ku.dk (YM)

**Competing interest:** The authors declare that no competing interests exist.

**Abstract** The glymphatic system is a fluid transport network of cerebrospinal fluid (CSF) entering the brain along arterial perivascular spaces, exchanging with interstitial fluid (ISF), ultimately establishing directional clearance of interstitial solutes. CSF transport is facilitated by the expression of aquaporin-4 (AQP4) water channels on the perivascular endfeet of astrocytes. Mice with genetic deletion of AQP4 (AQP4 KO) exhibit abnormalities in the brain structure and molecular water transport. Yet, no studies have systematically examined how these abnormalities in structure and water transport correlate with glymphatic function. Here, we used high-resolution 3D magnetic resonance (MR) non-contrast cisternography, diffusion-weighted MR imaging (MR-DWI) along with intravoxel-incoherent motion (IVIM) DWI, while evaluating glymphatic function using a standard dynamic contrast-enhanced MR imaging to better understand how water transport and glymphatic function is disrupted after genetic deletion of AQP4. AQP4 KO mice had larger interstitial spaces and total brain volumes resulting in higher water content and reduced CSF space volumes, despite similar CSF production rates and vascular density compared to wild-type mice. The larger interstitial fluid volume likely resulted in increased slow but not fast MR diffusion measures and coincided with reduced glymphatic influx. This markedly altered brain fluid transport in AQP4 KO mice may result from a reduction in glymphatic clearance, leading to enlargement and stagnation of fluid in the interstitial space. Overall, diffusion MR is a useful tool to evaluate glymphatic function and may serve as valuable translational biomarker to study glymphatics in human disease.

## Editor's evaluation

This important investigation is of interest to neuroimaging scientists and neurophysiologists studying the glymphatic system. Using a multi-modal approach including magnetic resonance and histological methods, this work provides substantial data interrogating the effect of removing aquaporin-4 (AQP4) from the mouse brain parenchyma on the structural morphology and interstitial fluid dynamics stagnation. In particular, the authors provide convincing evidence that deletion of AQP4 in mice results in increased interstitial volume, likely due to increased resistance to parenchymal CSF efflux.

## Introduction

Aquaporin (AQP) channels facillitate passive water transport across cell membranes (*Preston et al., 1992*; *Li and Wang, 2017*). Aquaporin-4 (AQP4) water channels are highly enriched in astrocytic endfeet that ensheath the cerebral vasculature (*Nagelhus and Ottersen, 2013*), covering up to 20–60% of the perivascular endfeet membrane facing the vessel wall (*Nielsen et al., 1997*; *Wolburg et al., 2011*; *Rash et al., 1998*; *Rasmussen et al., 2022*). This high density of water channels in astrocytic vascular endfeet is remarkable because brain endothelial cells are essentially devoid of the AQP1 water channels expressed by endothelial cells in peripheral tissues (*Bonomini and Rezzani, 2010*). AQP4 has primarily been studied in the context of pathophysiology, such as ischemia, traumatic brain injury, or hydrocephalus (*Iacovetta et al., 2012*; *Trillo-Contreras et al., 2018*; *Urushihata et al., 2021*; *Trillo-Contreras et al., 2021*; *Katada et al., 2014*; *Bloch et al., 2006*). Genetic deletion of AQP4 (AQP4 KO) in mice results in an increased brain water content (*Katada et al., 2014*; *Li et al., 2009*), an expanded interstitial fluid (ISF) volume fraction (*Katada et al., 2014*; *Yao et al., 2008*), decreased ventricular volume (*Trillo-Contreras et al., 2018*; *Li et al., 2009*), and decreased capacity to buffer interstitial potassium ($K^+$) ions (*Amiry-Moghaddam et al., 2003b*; *Strohschein et al., 2011*). Yet, why AQP4 was enriched at the endfeet of the astrocytes was unknown until the discovery of the glymphatic system (*Iliff et al., 2012*).

The glymphatic system is comprised of a network of annular perivascular spaces formed by astrocytic endfeet ensheathing the vascular walls. Perivascular spaces form a low resistance pathway enabling cerebrospinal fluid (CSF) and ISF exchange, promoting the clearance of interstitial solutes from the brain (*Iliff et al., 2012*). AQP4 KO mice exhibit a 25–60% decrease in glymphatic CSF tracer influx (*Mestre et al., 2018a*; *Hablitz et al., 2020*; *Zhang et al., 2019*), and acute pharmacological blockade of AQP4 inhibits glymphatic transport using TGN-020 inhibitor (*Huber et al., 2009*; *Harrison et al., 2020*; *Takano and Yamada, 2020*), reducing severity of brain edema and lesion volume after ischemic injury (*Igarashi et al., 2011*; *Sun et al., 2022*). However, in cell-based assays TGN-020 inhibitor failed to inhibit AQP4, bringing into question the true molecular mechanisms of its action (*Verkman et al., 2017*), discussed in *Choi et al., 2021*. Deletion of AQP4 also accelerates buildup of neurotoxic protein waste in neurodegenerative models of Alzheimer's (*Xu et al., 2015*; *Ishida et al., 2022*) and Parkinson's disease (*Cui et al., 2021*). In humans, a common single nucleotide AQP4 polymorphism is correlated to changes in slow-wave non-REM sleep and cognition (*Ulv Larsen et al., 2020*), consistent with increased glymphatic function during sleep (*Xie et al., 2013*; *Eide et al., 2021*). Recent reports highlight potential roles of AQP4, especially in edema formation after hypoxia due to the spinal cord injury, and in early and acute phases of stroke (*Salman et al., 2022*; *Kitchen et al., 2020*; *Sylvain et al., 2021*). Thus, the evidence suggests that it is the vascular polarized AQP4 expression in the astrocytic vascular endfeet functionally crucial for fluid transport, and not necessarily total AQP4 levels in the tissue (*Mestre et al., 2018a*; *Hablitz et al., 2020*; *Sylvain et al., 2021*; *Amiry-Moghaddam et al., 2003a*; *Eide and Hansson, 2018*).

Thus, AQP4 plays a strategic role in facilitating CSF influx across the vascular endfeet of astrocytes, waste clearance, and proper sleep architecture. However, it is not clear how AQP4 facilitates glymphatic fluid transport in part because of the lack of non-invasive whole-brain *in vivo* measurement of fluid dynamics.

The purpose of this study was to characterize the impact of genetic AQP4 deletion in mice on brain-water morphometry and transport by employing state-of-the-art multi-modal *in vivo* magnetic resonance imaging (MRI) alongside more traditional physiological and histological approaches to measure vascular density, distribution of AQP4 across the brain, brain-water content, ISF volume, and CSF production. Using fully non-invasive high-resolution 3D MR cisternography, we assessed structural differences between the intracranial and CSF space volumes of AQP4 KO and wildtype (WT) mice, as well as the water molecular diffusion and pseudodiffusion using standard diffusion-weighted imaging (DWI) and intravoxel-incoherent motion (IVIM) DWI. Finally, we superimposed these methods with standard dynamic contrast-enhanced MRI to generate the first comprehensive evaluation of brain-fluid movement in the AQP4 KO mouse model.

## Results

### AQP4 KO mice having larger brain volumes and smaller CSF spaces

First, we hypothesized that AQP4 deletion would alter macroscopic features of the adult murine brain as previously reported (*Trillo-Contreras et al., 2018*; *Katada et al., 2014*; *Li et al., 2009*; *Yao et al., 2008*; *Haj-Yasein et al., 2011*). We employed 3D constructive interference in steady-state (CISS)-based CSF space volumetry and cisternography to delineate highly $T_2$-weighted water signal allowing high-resolution mapping of the brain fluid compartments. We found the brain volume 5–10% larger in AQP4 KO than in WT mice (p<0.01, mean ± SD KO = 521 ± 13 vs. WT = 477 ± 24 mm³; *Figure 1A*) with no significant differences between animal age (p=0.697), body weight (p=0.7662) or signal-to-noise ratio (p=0.1385) of the 3D-CISS images from two genotypes (*Table 1– Methods*). This difference in brain volume coincided with ~6% increase in the brain water content measured *ex vivo* (p<0.05, min-max of 2–11%, KO = 3.66 ± 0.09 vs. WT = 3.46 ± 0.05 ml/g dry brain weight; *Figure 1D*; *Li et al., 2009*; *Haj-Yasein et al., 2011*).

Total CSF volume was estimated as 2.4–4.4% of the parenchymal volume among all animals and no difference in delineated CSF space volumes was found between KO and WT (p=0.1255, KO = 15.1 ± 1.9 vs. WT = 17.7 ± 2.4 mm³). Yet the whole segmented CSF space to brain volume ratio was 23–29% smaller in KO compared to WT (p<0.05, KO = 2.99 ± 0.43 % vs. WT = 3.86 ± 0.41 %; *Figure 1B*). This difference was mainly noted within the ventricular system comprising of the lateral, third and fourth ventricles (p<0.05; *Figure 1C*), consistent with previous *ex vivo* (*Li et al., 2009*) or lower resolution 2D assessment with low-field strength MRI (*Trillo-Contreras et al., 2018*; *Trillo-Contreras et al., 2021*; *Li et al., 2009*). The most prominent difference was in the lateral (p<0.05, mean ± SEM WT-KO = 0.57 ± 0.15 %) and third ventricles (p<0.01, WT-KO = 0.13 ± 0.03 %), but not the fourth ventricle (p=0.6623; *Appendix 1—figure 1A*). These changes in the ventricular volume were not driven by CSF production, since we found similar CSF volume production in AQP4 KO and WT mice (*Figure 1E*) using a newly developed *in vivo* approach (*Liu et al., 2020*). No difference in the CSF spaces of the parietal cisterns was noted (p=0.4589; *Appendix 1—figure 1A*). Perhaps most surprisingly, no differences were noted in the segmented perivascular CSF space between KO and WT (p=0.1623; *Figure 1C*) or its individual components (skull base/ Circle of Willis, p=0.9307; basilar artery, p=0.4286; anterior/posterior perivascular spaces, p=0.2486; *Appendix 1—figure 1A*). Quantification of interstitial space volume using real-time iontophoresis with tetramethylammonium (TMA) (*Odackal et al., 2017*), showed that both awake and ketamine/xylazine anesthetized AQP4 KO mice exhibited a larger interstitial space (p<0.05 for both, *Figure 1F*). The relative enlargement in the interstitial space volume fraction, $\alpha$, that occured in response to ketamine/xylazine administration did not differ between the two genotypes (p=0.9186, $\Delta\alpha_{\text{awake-K/X}}$ KO = 0.090 ± 0.047 vs. WT = 0.093 ± 0.059; Mann-Whitney U-test). We found no difference in tortuosity between genotypes (p=0.1412, *Figure 1F*). Thus, deletion of AQP4 is linked to an expansion of the interstitial space volume fraction as well as in total brain volume, with no clear abnormalities in the glymphatic influx paths such as the size of the larger periarterial spaces.

### Genetic loss of AQP4 alters water diffusivity independent of the microvascular density

We next asked how deletion of AQP4 affected the brain's water mobility within the brain parenchyma and CSF compartments. First, we used a standard DWI model which, by assuming monoexponential signal decay using apparent diffusion coefficient (ADC), provides very sensitive but non-specific scoring for cellularity, the integrity of the cell membranes, and difference in intracellular and ISF volumes (*Le Bihan et al., 1988*) or their composition and viscosity (*Le Bihan et al., 1986*; *Le Bihan and Iima, 2015*). Second, we applied a biexponential intravoxel-incoherent motion (IVIM) DWI model (*Le Bihan et al., 1988*; *Le Bihan et al., 1986*) to measure passive molecular water diffusion (D) separately from the water motion affected by tissue perfusion (*Fournet et al., 2017*; *Federau, 2017*; *Vieni et al., 2020*).

ADC and D (IVIM) provided similar results (Pearson's linear correlation, r=0.94, *P*<0.0001, *Appendix 1—figure 1B*), and no differences were found within all 5 large CSF space regions (*Table 2*, *Appendix 1—figure 1C*). Both revealed increased slow diffusion measures within the brain parenchyma, with KO animals exhibiting ADC and D 5.7 ± 1.5 % higher than in WT (*Figure 2A and C*), consistent with previous ADC estimates in KO animals using lower resolution at 7 Tesla MR and

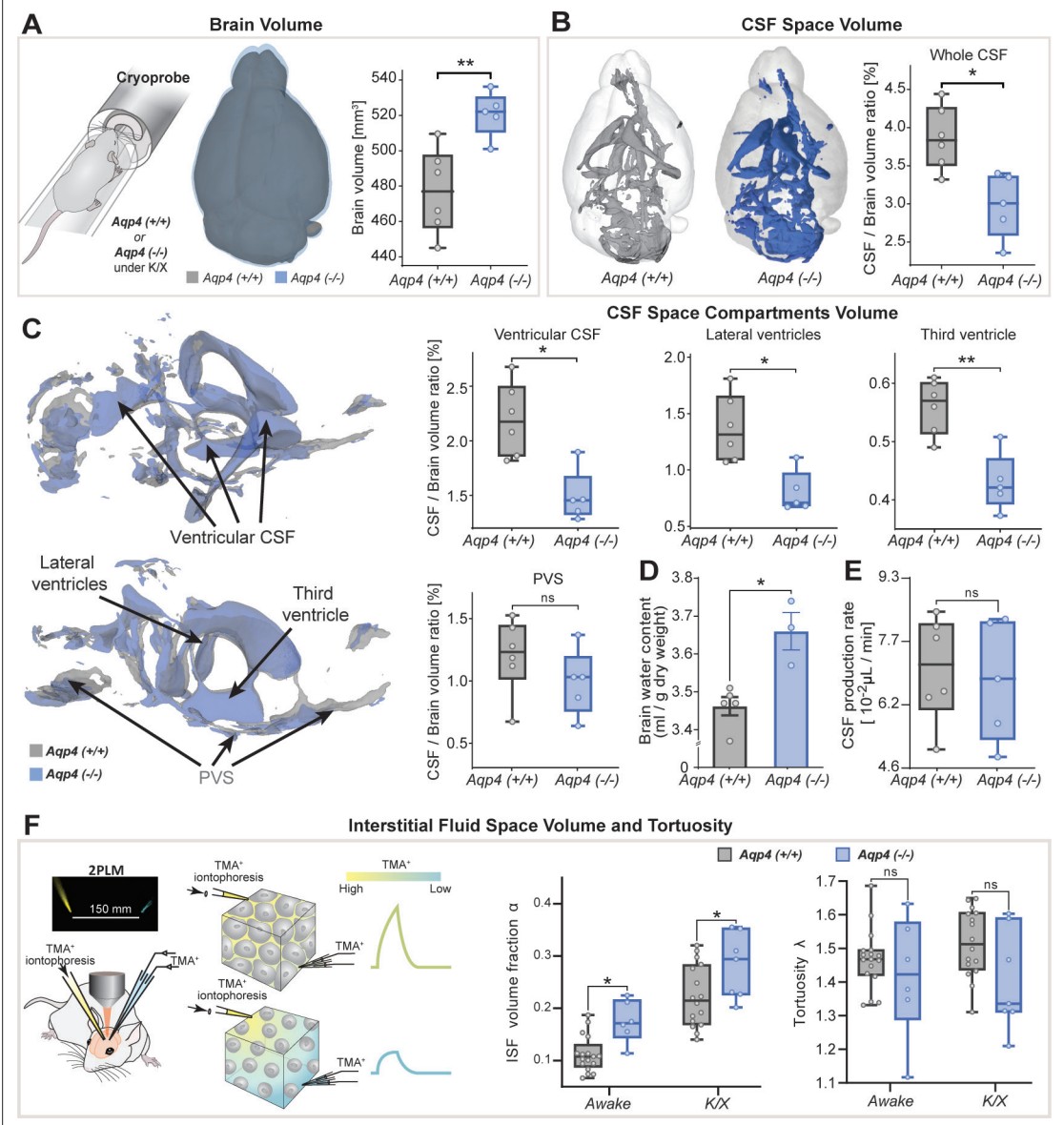

**Figure 1.** 3D-CISS MRI, CSF and interstitial space volumetry *in vivo*. Overlaid 3D surface images of the co-registered and averaged 3D-CISS brain volumes, and whiskers-box plots comparing (**A**) the brain volumes and (**B**) segmented CSF space volumes from 6 WT and 5 AQP4 KO mice (see *Figure 1—source data 1*). (**C**) Overlaid 3D surface reconstruction of the co-registered whole CSF spaces segmented from 3D-CISS from all WT (gray) and KO (blue) animals, along with whiskers-box plot comparisons of main CSF compartments volumes segmented: whole ventricular (**left top**), lateral (**middle top**) and third ventricular (**right top**), and whole perivascular space at the skull base (PVS; **left bottom**). (**D**) Whiskers-box plots comparing brain water content from 5 WT and 3 KO *ex vivo* (*Figure 1—source data 2*). (**E**) Whiskers-box plots comparing CSF production rates measured in 6 WT and 5 KO *in vivo* (*Figure 1—source data 3*). (**F**) Whiskers-box plots for the extracellular space volume and tortuosity measured using real-time iontophoresis with tetramethylammonium (TMA) in awake (17 WT, 6 KO) and K/X anesthetized (16 WT, 7 KO) animals (*Figure 1—source data 4*). **Legend:** ns-not significant, *-p<0.05, **-p<0.01; Mann-Whitney U-test (**A–E**), one-way ANOVA with Bonferroni's post-hoc correction (**F**).

The online version of this article includes the following source data for figure 1:

**Source data 1.** 3D-CISS CSF space volumetry *in vivo* - source data.

**Source data 2.** Brain water content *ex vivo* - source data.

**Source data 3.** CSF production *in vivo* - source data.

**Source data 4.** ISF space volume estimation with TMA - source data.

**Table 1.** Summary of demographic characteristics of the animals used in 8 experiments performed, and (B) details of MRI protocols and acquisition parameters for evaluation of the brain CSF and ISF spaces, employed in the current study.

Legend: CryoProbe – cryogenically-cooled MR coil; TA - time of acquisition; Tx/Rx - transmit/receive; TR - time to repetition; TE - time to echo; FA - flip angle; FOV - field of view; Av. / Rep. – averages for image formation or independently repeated acquisitions; FISP - steady-state free precession sequence; TrueFISP - true balanced steady-state free precession sequence; EPI - echo-planar imaging sequence; VTR – variable TR; VFA – variable flip angle; CM – cisterna magna; $p$ - statistical p-value from Mann-Whitney U-test; N/A – data not available; NS - statistically not significant finding ($p>>0.05$).

**A**

| | No.of animals | | | Age [weeks] | | Body weight [g] | | Respiration [bpm] | |
|---|---|---|---|---|---|---|---|---|---|
| | KO | WT | Male [%] | Overall (mean ±SD) | p | Overall (mean ±SD) | p | Overall (mean ±SD) | p |
| 3D-CISS volumetry | 5 | 6 | 72.7 | 13.8±1.8 | NS | 27.3±2.1 | NS | 176±30 | NS |
| MR-DWI | 6 | 6 | 41.7 | 10.4±0.7 | NS | 22.4±3.0 | NS | 175±11 | NS |
| DCE-MRI | 5 | 6 | 45.5 | 13.4±1.6 | NS | 27.3±2.2 | NS | 182±14 | NS |
| AQP4 expression | - | 4 | 50 | 10.0±0.0 | NS | 25–30 | N/A | - | - |
| Vascular density | 6 | 6 | 0 | 14.6±1.0 | NS | 24.3±4.6 | NS | - | - |
| CSF production | 5 | 6 | 45.5 | 15.4±0.5 | NS | 28.7±4.2 | NS | - | - |
| ISF space volume estimation using TMA | 8 | 20 | 55 | 10.0±0.0 | NS | 25–30 | N/A | - | - |
| Brain water content | 3 | 5 | 62.5 | 12.0±1.0 | NS | 25–30 | N/A | - | - |

**B** MR sequences and acquisition parameters

| Sequence (Tx/Rx coil; slice orientation) | TR [ms] | TE [ms] | FA [deg] | Av./Rep. | Voxel size [mm³](interpolation) | Bandwidth [Hz/pix] | Matrix size | TA |
|---|---|---|---|---|---|---|---|---|
| **MR CSF space volumetry** | | | | | | | | |
| 3D-TrueFISP (CryoProbe; sagittal) | 5.2 | 2.6 | 50 | 2 | 0.033×0.033×0.033 (2.0×1.6×1.0) | 260 | 19.2×12.8×12.8 | 27 min |
| MR-DWI (in vivo and ex vivo) (δ=3 and Δ=10ms for gradient duration and separation times) *(b-values (Av.>1): 40, 50, 59, 70, 92, 113, 165, 197, 238, 342, 445, 649, 854, 1057 (2), 1564 (2), 2071 (2), 3081 (2) s²/mm) 2D-EPI (volumetric; axial) | 3600 | 30 | 90 | 3 | 0.15×0.15×0.5 (0.2 mm gap, 16 slices) | 3307 | 16.2×14.4×11.2 | 20–30 min (respiratory-gated) |
| **DCE-MRI via CM-injection** 3D-FISP (CryoProbe; sagittal) | 3.26 | 1.63 | 15 | 1 | 0.1×0.1×0.1 | 781 | 19.2×12.8×12.8 | 90 min |
| **Microbeads phantom ex vivo - T1 mapping** (VTR: 12000, 9000, 6500, 4000, 2000, 1000, 800, 500, 300, 100, 80, 50, 15ms) (VFA: 45° and 90°) 2D-RARE (volumetric; axial) | VTR 12000 | 3.1 | 90 | 2 | 0.1×0.1×3.0 | 671 | 16.2×16.2×1 | 3 h 20 min |
| 2D-RARE (volumetric; axial) | VTR 12000 | 3.1 | VFA | 3 | 0.1×0.1×3.0 | 671 | 16.2×16.2×1 | 1 hr 40 min |

*Presented averages of measured values from 3 diffusion encoding directions are slightly higher than the set-up, due to gradient preparation time.

**Table 2.** Summary of findings for (A) average ADC and D-IVIM, (B) direction-wise MR diffusion and pseudodiffusion among 21 ROI assessed, along with statistical scoring.

Asterisks reflect 'p' significance values from the nonparametric Mann-Whitney test comparing diffusion measures between KO vs. WT animals ROI-wise (total n=21, balanced groups), along with the magnitude of the difference expressed by the inequality sign. Legend: OLF-olfactory, CA / RSP-cingulate / retrosplenial, VIS-visual (V1), SS-somatosensory (S1), AUD-auditory, HIP-hippocampus, PERI-perirhinal, TH-thalamus, HAB-habenula, HY-hypothalamus, MB-midbrain, PAG-periaqueductal gray, HB-hindbrain; CP-caudate putamen, WM-white matter; 3V-third ventricle, LV-lateral ventricle, 4V-fourth ventricle, PCS-pericisternal space, CoW-Circle of Willis, CB-cerebellum. *NS*- no significant difference, *-p<0.05, **-p<0.01, by means of Mann-Whitney U-test.

| A | | Average ADC | | Average D-IVIM | | IVIM |
|---|---|---|---|---|---|---|
| | ROI | Finding | Significance | Finding | Significance | *D\*, Fp, Fp x D\** |
| Cerebral cortex | OLF | KO>WT | * | - | NS | - |
| | CA / RSP | - | NS | - | NS | - |
| | VIS | - | NS | - | NS | - |
| | SS | KO>WT | * | KO>WT | * | - |
| | AUD | KO>WT | * | KO>WT | * | - |
| | HIP | KO>WT | ** | KO>WT | * | - |
| | PERI | - | NS | - | NS | - |
| Brain stem | TH | KO>WT | ** | KO>WT | ** | - |
| | HAB | - | NS | - | NS | - |
| | HY | - | NS | - | NS | - |
| | MB | KO>WT | * | KO>WT | * | - |
| | PAG | KO>WT | * | - | NS | - |
| | HB | KO>WT | ** | KO>WT | ** | - |
| Cerebral nuclei and tracts | CP | KO>WT | ** | KO>WT | * | - |
| | WM | KO>WT | * | KO>WT | * | - |
| CSF space | 3V | - | NS | - | NS | - |
| | LV | - | NS | - | NS | - |
| | 4V | - | NS | - | NS | - |
| | PCS | - | NS | - | NS | - |
| | CoW | - | NS | - | NS | - |
| Cerebellum | CB | - | NS | - | NS | KO>WT Fp = 0.0649 |

*Table 2 continued on next page*

*Table 2 continued*

| B | ROI | IVIM | | | | | | ADC / D | | | | | |
|---|---|---|---|---|---|---|---|---|---|---|---|---|---|
| | | Direction Z (cranio-caudal) | | Direction X (bilateral) | | Direction Y (ventral-dorsal) | | Direction Z (cranio-caudal) | | Direction X (bilateral) | | Direction Y (ventral-dorsal) | |
| | | Finding | Signif. | Finding | Signif. | Finding | Signif. | Finding | Signif. | Finding | Signif. | Finding | Signif. |
| Cerebral cortex | OLF | - | NS | Fp x D* KO>WT | * | - | NS | KO>WT | **/* | - | NS | - | NS |
| | CA / RSP | D* KO>WT | * | - | NS | - | NS | - | NS | - | NS | - | NS |
| | VIS | - | NS | - | NS | - | NS | - | NS | KO>WT | */- | KO>WT | */- |
| | SS | - | NS | - | NS | - | NS | KO>WT | -/* | KO>WT | **/ P=0.056 | KO>WT | */- |
| | AUD | - | NS | - | NS | - | NS | KO>WT | */* | KO>WT | **/* | KO>WT | */* |
| | HIP | - | NS | - | NS | - | NS | KO>WT | */* | KO>WT | */- | KO>WT | **/* |
| | PERI | D* KO>WT | ** | - | NS | - | NS | - | NS | KO>WT | */- | - | NS |
| Brain stem | TH | - | NS | - | NS | - | NS | KO>WT | **/** | KO>WT | */- | KO>WT | */- |
| | HAB | - | NS | - | NS | - | NS | - | NS | - | NS | - | NS |
| | HY | - | NS | - | NS | - | NS | - | NS | - | NS | - | NS |
| | MB | - | NS | - | NS | - | NS | KO>WT | -/* | KO>WT | */- | KO>WT | **/- |
| | PAG | - | NS | - | NS | - | NS | KO>WT | */** | KO>WT | **/- | KO>WT | */- |
| | HB | - | NS | - | NS | - | NS | KO>WT | */* | KO>WT | **/- | KO >WT | */* |
| Cerebral nuclei and tracts | CP | - | NS | - | NS | - | NS | KO>WT | **/** | KO>WT | */- | - | NS |
| | WM | D* KO>WT | ** | - | NS | - | NS | - | NS | - | NS | KO>WT | */- |
| CSF space | 3V | - | NS | Fp / Fp x D* KO>WT | ** / * | - | NS | - | NS | - | NS | - | NS |
| | LV | - | NS | - | NS | - | NS | - | NS | - | NS | - | NS |
| | 4V | - | NS | - | NS | - | NS | - | NS | - | NS | - | NS |
| | PCS | - | NS | - | NS | - | NS | - | NS | - | NS | - | NS |
| | CoW | - | NS | - | NS | - | NS | - | NS | - | NS | - | NS |
| Cerebellum | CB | - | NS | - | NS | - | NS | - | NS | - | NS | - | NS |

time-dependent diffusion (*Pavlin et al., 2017*). This was evident in 10 out of 15 parenchymal regions assessed (*Figure 2A*), with the largest differences visible in 4 brainstem areas (thalamic, midbrain, periaquaductal gray, and hindbrain) as well as 4 cortical regions (olfactory, somatosensory, auditory, and hippocampal), and the caudate region (min. p<0.05 for all). Overall, these results support the hypothesis of increased ISF space volume, with no difference in slow water mobility within the large CSF spaces in KO animals.

To evaluate whether a fast bulk displacement of intravascular water protons due to capillary perfusion may contribute to our findings, we performed an additional scoring for differences associated with intra-voxel pseudodiffusive fluid regimes (*Le Bihan and Iima, 2015*), using IVIM diffusion model. IVIM may reveal pathophysiological impairment in the microcirculation by estimating perfusion fraction ($F_p$) and pseudodiffusion coefficient (fast diffusion, D*; *Le Bihan and Turner, 1992*; *Henkelman, 1990*; *Henkelman et al., 1994*), but can also detect general fluid dynamics associated with macromolecules such as proteins or biological polymers (*Le Bihan, 2019*). Transferring IVIM measures to standard perfusion measurements, D* can be associated with mean transit time, $F_p$ with a flowing blood fraction that is correlated with vessel density or cerebral blood volume, and the product of $F_p \times D*$ with relative cerebral blood flow in each voxel. We found no differences in average D*, $F_p$ and the product of

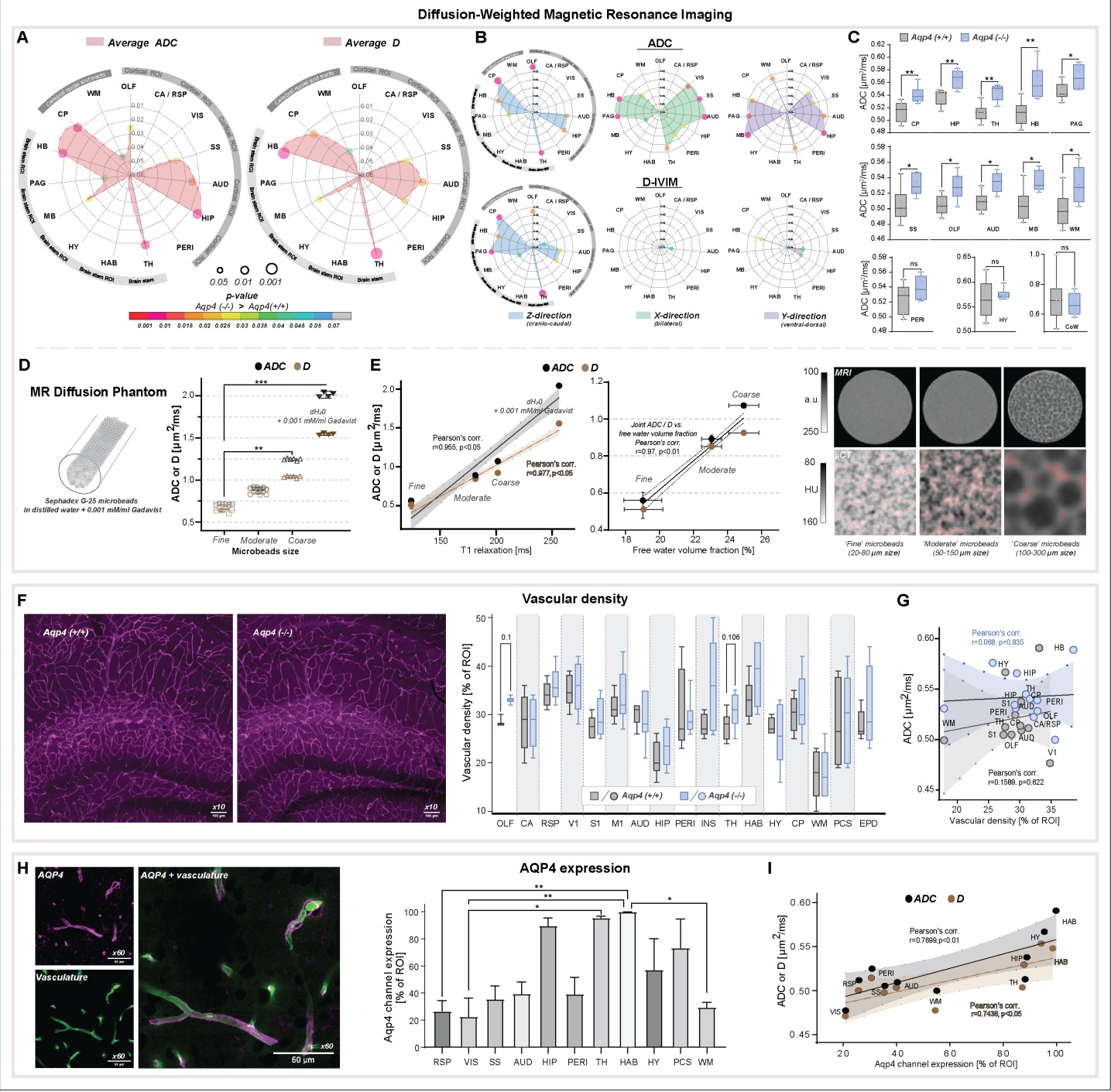

**Figure 2.** Diffusion-weighted MRI *in vivo* and *ex vivo*, vascularity, and AQP4 cellular surface expression *ex vivo*. Radar plots showing statistical significances for the differences between the slow diffusion measures among 15 parenchymal ROI assessed, for average ADC (**A**) and D, and in (**B**) 3 diffusion-encoding direction separately; (**C**) Whiskers-box plots for the mean of the average ADC values among 6 AQP4 KO and 6 WT animals analyzed, including: the regions showing the most significant differences (**top**); exemplary regions showing significant differences (**middle**); exemplary regions showing no differences (**bottom**), by means of the Mann-Whitney U-test (*Figure 2—source data 1*); (**D**) mean and 95% confidence intervals of ADC and D calculated in a water phantom (+0.001 mM/ml gadobutrol) and 3 water phantoms filled with Sephadex-G25 microbeads of fine, moderate and coarse sizes; (**E**) Correlation plots of the calculated mean ± SD of ADC and D to the $T_1$ relaxation values and free fluid volumes obtained from the phantoms using MRI (**left**), and micro-computed tomography (μCT; **middle**) (*Figure 2—source data 2*). Single-slices of turbo-spin echo (RARE) MR images from respective phantoms (**right, upper**) along with μCT images of the central portion of the respective phantoms (**right, lower**) filled with 1:1 solution of Ominpaque 350 contrast agent and 0.9% NaCl. Semi-transparent red area marks the free fluid space considering only voxels above 75th percentile

*Figure 2 continued on next page*

*Figure 2 continued*

of Hounsfield units (HU) intensity distribution in each μCT image; (**F**) Exemplary immunohistochemistry images (Olympus UplanXApo 10 x/numerical aperture 0.40, ∞/compatible cover glass thickness 0.17 mm/field number 26.5 mm, no immersion liquid) (**left**) from the hippocampal area from WT and KO animal (magenta vascular labeling) along with whiskers-box plot for the comparison of the mean vascular density among 17 regions analyzed (**right**) (*Figure 2—source data 3*); (**G**) Region-wise correlation plot of calculated ADC to vascular density among 12 regions analyzed with both methods, in WT and KO animals; (**H**) Exemplary image for vasculature (green, AlexaFluor 488) and AQP4 (magenta) immunohistochemistry staining (Olympus UplanXApo 60 x/numerical aperture 1.42, ∞/compatible cover glass thickness 0.17 mm/field number 26.5 mm, oil immersion) (**left**), bar-plot comparison of the mean AQP4 channel expression among 11 ROI assessed from 4 WT mice (**right**) (*Figure 2—source data 4*); (**I**) ROI-wise correlation plot comparing average ADC and D with the mean AQP4 channel expression among 10 regions assessed with both methods. **Legend:** OLF-olfactory, CA / RSP-cingulate / retrosplenial, VIS (**V1**)-visual, SS (**S1**)-somatosensory, M1-motorcortex, AUD-auditory, HIP-hippocampus, PERI-perirhinal, INS-insular, TH-thalamus, HAB-habenula, HY-hypothalamus, MB-midbrain, PAG-periaqueductal gray, HB-hindbrain; CP-caudate putamen, WM-white matter; PCS-pericisternal space, CoW-Circle of Willis, EPD-ependymal layer around lateral ventricles; SD-standard deviation; ns-not significant, *-p<0.05, **-p<0.01, by means of Mann-Whitney U-test (**A–C, F**), Kruskal-Wallis one-way ANOVA with Dunn's correction (**D, H**). All correlation plots show respective regression lines along with semi-transparent areas marking 95% confidence intervals of the fitting. The highest obtained Pearson's linear or Spearman's range correlation scores are reported and considered significant if correlation value >0.5 with p<0.05, and non-zero regression slope.

The online version of this article includes the following source data for figure 2:

**Source data 1.** DWI and IVIM-DWI estimates *in vivo* - source data.

**Source data 2.** DWI, IVIM-DWI, and T1 estimates in the phantom - source data.

**Source data 3.** Immunohistochemistry: vascular labeling - source data.

**Source data 4.** Immunohistochemistry: AQP4 channel labeling - source data.

$F_p \times D^*$ within all parenchymal regions from KO and WT mice (*Table 2A*). This suggests similar blood perfusion, and is supported with our histological analysis showing no significant differences in the vascular density between the genotypes (*Figure 2F*), with a trend towards a small increase within the thalamus and olfactory bulb (11 and 15 %, respectively; both p≈0.1). The lack of change in both IVIM measures and vessel density supports the conclusion that no tangible differences in microcirculation exist between KO and WT and is supported by no difference in the $H_2^{17}O$ washout between KO and WT at 9.4 Tesla MR (*Zhang et al., 2019*).

## Directional water diffusion as a measure of anatomical differences in AQP4 KO mice

Application of both DWI and IVIM models confirmed increased slow MR diffusion in the brain parenchyma, with no difference in the fast MR diffusion (psuedodiffusion) in AQP4 KO mice (*Table 2A*, *Appendix 1—figure 1B–C*). We next investigated tissue orientation-specific water mobility restrictions, by assessing the diffusion separately for cranio-caudal (slice, Z), in-plane bilateral (X), and ventral-dorsal (Y) encoding directions. Overall, the largest differences in both ADC and D were found in the direction parallel to the main orientation of neuronal tracts (i.e. bilateral for the auditory cortex, bilateral and ventral-dorsal for hindbrain, ventral-dorsal for midbrain, cranio-caudal for caudate and thalamus).

ADC revealed a similar degree of increased water diffusion in KO compared with WT, among 3 brain stem regions (thalamus, periaqueductal gray, hindbrain; min. p=0.04; *Figure 2B*) and 4 cortical regions (visual, somatosensory, auditory, hippocampal; min. p<0.03) in both bilateral and ventral-dorsal directions. D highlighted the main differences present only in the cortical somatosensory, auditory, and hippocampal regions (*Figure 2A* vs. *Figure 2B*). In the cranio-caudal direction, ADC model highlighted main differences between the genotypes in the same 3 brain stem regions (*Figure 2B*) and 3 cortical regions (olfactory, auditory, hippocampal), and the largest difference was visible in the olfactory, thalamic, and the caudate regions (p<0.01). In this direction, D had higher sensitivity (*Figure 2B*) and the largest difference in D appeared in the parenchymal brain stem areas neighboring the ventricular spaces (i.e. thalamus and periaqueductal gray; p<0.01), which possibly reflects the disrupted ependymal cell layer around ventricles and cerebral aqueduct in AQP4 KO mice (*Li et al., 2009*).

Fast water diffusion was altered between KO and WT solely in the cranio-caudal (Z) and bilateral (X) directions (*Appendix 1—figure 1C*), and were usually associated with a difference in slow diffusion markers in at least one orthogonal direction (*Table 2*). D* was different between the genotypes in the cranio-caudal direction within cingulate/retrosplenial and perirhinal cortex, and white matter (p<0.01, *Table 2B*). This might reflect possible differences in the rate of water passage orthogonally

to the differences in associated fluid perfusion markers. Differences in $F_p$ and $F_p \times D^*$ were found only in the olfactory area and within the 3rd ventricle (min. p<0.05; *Table 2B*), suggesting respectively possible alterations in interstitial fluid efflux pathways and in CSF passage within the smaller volume of 3rd ventricle in KO.

Overall, slightly increased average ADC and D along with no difference in average IVIM measures suggest existence of larger interstitial space volume in AQP4 KO, perhaps as a result of increased water exchange time (*Urushihata et al., 2021*), and without tangible alterations in parenchymal blood perfusion. We tested this hypothesis by mimicking increasing interstitial space volume using three water phantoms filled with Sephadex G-25 microbeads of coarse, moderate and fine particle size (*Figure 2D and E*). The coarse beads will have greater spaces between the particles than the fine (see MRI and μCT images in *Figure 2E*), yet all microbeads possess the same porosity so a similar exchange rate between stored and free water pools is expected. We found both ADC and D increased along with the free fluid space volume surrounding the microbeads, as expected (*Lee et al., 2016*), with no difference in IVIM measures between the phantoms with microbeads (*Appendix 1—figure 1E*). This was reflected with high linear correlation between ADC and D, and $T_1$ relaxation times (r=0.977, p<0.05) and free fluid volume fractions estimated (r=0.97, p<0.01) using MRI and μCT in all phantoms (*Figure 2D-E*). We conclude that these increased ADC and D with increased free water pools support larger interstitial space volume in AQP4 KO compared to WT mice.

## Reduced gadolinium-based MRI tracer influx into AQP4 KO mouse brain

Our noninvasive MR measurements showed increased ADC values along with increased brain volume and reduced CSF space in AQP4 KO mice. Next, we tested whether these measures were associated with reduction in the gadolinium CSF tracer influx from the cisterna magna into the brain parenchyma by means of standard DCE-MRI (*Figure 3A*). As of particular importance for studying AQP4, it is worth noting that tracer transport (here gadobutrol) does not directly reflect the movement of water. The water can move into the tissue not only through the paracellular gap between astrocytic endfeet but also via diffusive transcellular exchange. The transport of membrane-impermeable CSF tracers, however, is limited to paracellular transport between the gaps of astrocytic endfeet (*Salman et al., 2022*; *Salman et al., 2021*).

In contrast to prior report (*Li et al., 2009*), but consistent with the glymphatic model where ventricular fluid dynamics are upstream of cisterna magna injections (*Iliff et al., 2012*), no differences in the tracer distribution were found in the ventricular systems. From the cisterna magna (*Figure 3A*), the CSF tracer dispersed via the subarachnoid space cisterns to the Circle of Willis, and then dorsally along the middle cerebral artery into the brain parenchyma and anteriorly toward the olfactory bulb (*Figure 3B*), consistent with the previous reports using fluorescent tracers (*Iliff et al., 2012*; *Mestre et al., 2018a*). Importantly, there were no differences in the tracer distribution within the perivascular space at the basal cistern (*Figure 3B* – 'Circle of Willis'), consistent with the perivascular space volume not differing between the two genotypes (*Figure 1C*, *Appendix 1—figure 1A*). Yet, the peak and overall magnitude of the parenchymal signal enhancement were significantly lower in KO than WT brains (lowest p<0.01; *Figure 3B–D*). Differences were especially visible in the striatum (p<0.01), thalamus (p<0.05), hippocampus (p<0.01), and visual and cingulate/retrosplenial cortex (p<0.01 for both; *Figure 3C*). Within parenchyma, the difference increased with time from infusion and was largest in the cortex and hippocampus (p<0.001 at T=80 min; *Figure 3C and E*, and *Appendix 1—figure 2A*). Finally, while the tracer in WT accumulated around the venous sinuses as previously reported (*Iliff et al., 2013*), the AQP4 KO mice exhibited substantially less accumulation (p=0.0148; *Figure 3D*). The reduced parenchymal influx of contrast agent after cisterna magna injection in KO vs. WT mice are consistent with current models of glymphatic function (*Iliff et al., 2012*; *Mestre et al., 2018a*; *Kress et al., 2014*; *Mestre et al., 2018b*).

Next, to identify possible differences in the tracer dynamics between KO and WT mice, we calculated area under curve, arrival time, time-to-peak, peak intensity, and duration of significant from baseline parenchymal tracer accumulation (*Table 3*). Overall, the arrival time was similar, if slightly longer in KO than in WT mice. The largest delay in the tracer arrival time was visible in the lateral ventricles and caudate nucleus in KO (*Table 3*). The duration of time-to-peak was ~30% longer in KO among all regions, with the largest difference visible in the hippocampus, midbrain,

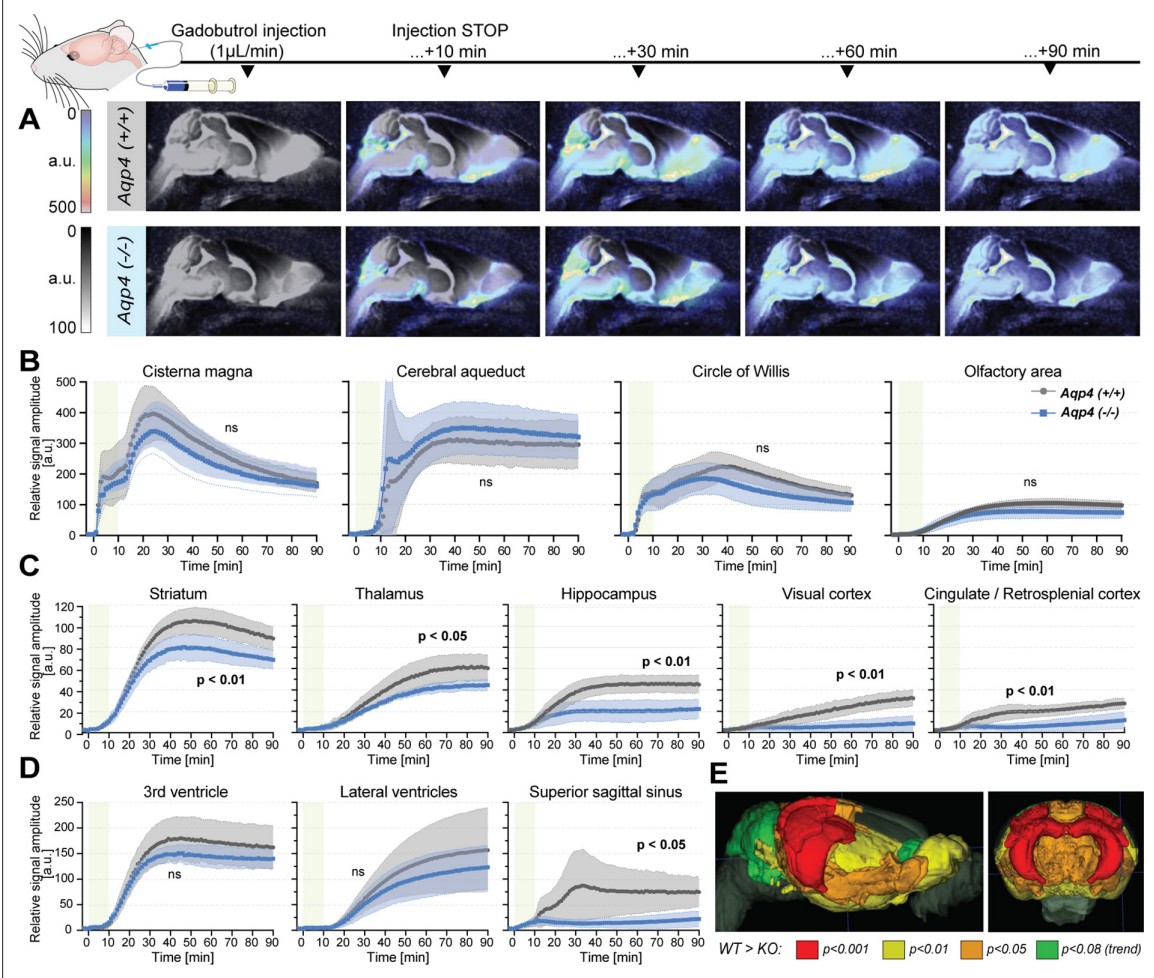

**Figure 3.** Dynamic contrast-enhanced MRI *in vivo* (*Figure 3—source data 1*). (**A**) 3D multiplanar reconstructions of dynamic-contract enhanced (DCE) MRI – sagittal slices from mean images of 6 WT (top) and 5 AQP4 KO (bottom) using 3D-FISP, after applying gadobutrol injection via cisterna magna (CM); mean ± SD DCE time-curves from WT (gray) and AQP4 KO (blue): (**B**) main CSF compartments ventrally and caudo-cranially from CM, (**C**) parenchymal regions where significant differences between WT and KO were found, (**D**) ventricular and CSF efflux regions. (**E**) 3D multiplanar reconstruction of DCE-MRI from mean 3D FISP images of 6 WT animals, along with Allen Brain Atlas-based segmentation maps color-coded according to p-significance value from nonparametric Two-way ANOVA with post-hoc, showing WT vs. KO differences in the CSF tracer dynamics at 60th minute after gadobutrol injection start. Legend: ns-not significant, *-p<0.05, **-p<0.01, by means of nonparametric Two-way Anova with Bonferroni's post-hoc.

The online version of this article includes the following source data for figure 3:

**Source data 1.** DCE-MRI *in vivo* - source data.

thalamus, cerebellum and superior sagittal sinus, though this trend was not significant due to high variability. In regions neighboring or related to the CSF spaces and dorsal cortex (i.e. olfactory and somatosensory cortex, hypothalamus, periaqueductal gray, hindbrain or perivascular) time-to-peak was longer in WT, which might be an overall effect of lower tracer penetration in KO. The relative WT-KO differences of peak intensities and parenchymal accumulation durations were always positive and moderately correlated (Pearson's linear correlation of relative peak intensity to relative duration difference of r=0.7212, p=0.0003; *Appendix 1—figure 2B*), which suggests that duration was shorter in KO due to smaller tracer penetration into the brain. This would be confirmed with the area under the DCE curve that consistently was smaller in KO (*Table 3*, **'Mean AUC'**). Together, delayed tracer arrival time and time-to-peak, lower peak intensity, and duration of tracer accumulation, as well as smaller area under the DCE curve in KO mice indicate a reduced tracer parenchymal influx and higher parenchymal resistance associated with lack of AQP4 channels (*Mestre et al., 2018a*).

**Table 3.** DCE-derived parameters from 21 ROI (matching those analyzed by means of MR-DWI) in 5 KO and 6 WT littermate mice, along with associated nonparametric pair-wise statistics using a two-tailed Wilcoxon signed-rank test and median ± standard deviation (SD) values strain-wise.

Legend: Mean AUC – mean from areas under the DCE curves along with associated p-statistical values (p-stat) from nonparametric Mann–Whitney U-test ROI-wise; Duration – duration of significantly different from the baseline signal enhancement, mimicking the parenchymal tracer accumulation; Aqp4(+/+) – wildtype control; Aqp4(-/-) – AQP4 KO mice; Δ – WT-KO difference; rel. Δ – ((WT-KO)/WT)×100% relative difference (negative '-' sign means the shorter duration of interstitial tracer accumulation in KO mice); For ROI abbreviations, see caption *Figures 2 and 3*, or *Table 2*. # –Two-way Anova with Bonferroni's post-hoc. ¶ – mean of standard deviations ROI wise. * – p<0.05, ** – p<0.01.

| ROI | | "WT vs. KO time-series different? | Mean AUC [a.u.] | | | Arrival time [min] | | | Time- to-peak [min] | | | Peak intensity [a.u.] | | | Interstitial tracer accumulation time [min] | | | | Duration [min] | | |
|---|---|---|---|---|---|---|---|---|---|---|---|---|---|---|---|---|---|---|---|---|---|
| | | | Aqp4 (+/+) | Aqp4 (-/-) | p-stat | Aqp4 (+/+) | Aqp4 (-/-) | Δ | Aqp4 (+/+) | Aqp4 (-/-) | Δ | Aqp4 (+/+) | Aqp4 (-/-) | rel. Δ [%] | Aqp4(+/+) Start | End | Aqp4(-/-) Start | End | Aqp4 (+/+) | Aqp4 (-/-) | rel. Δ [%] |
| Cortical | OLF | NS | 7207 | 5619 | NS | 6 | 6 | 0 | 61 | 52 | 9 | 106 | 79 | 25.5 | 43 | 80 | 39 | 70 | 38 | 32 | -15.8 |
| | CA/RSP | ** | 1662 | 616 | ** | 7 | 7 | 0 | 89 | 90 | -1 | 27 | 12 | 55.6 | 59 | 90 | 66 | 90 | 32 | 25 | -21.9 |
| | VIS | ** | 1699 | 529 | ** | 7 | 8 | -1 | 89 | 90 | -1 | 33 | 9 | 72.7 | 54 | 90 | 73 | 90 | 37 | 18 | -51.4 |
| | SS | NS | 537 | 438 | NS | 6 | 7 | -1 | 20 | 16 | 4 | 8 | 7 | 12.5 | 12 | 32 | 11 | 25 | 21 | 15 | -28.6 |
| | AUD | NS | 454 | 340 | NS | 7 | 7 | 0 | 89 | 89 | 0 | 10 | 5 | 50.0 | 67 | 90 | N/A | N/A | 24 | N/A | <-100 |
| | HIP | ** | 3258 | 1622 | * | 5.5 | 7.5 | -2 | 63 | 81 | -18 | 46 | 22 | 52.2 | 45 | 90 | 74 | 90 | 46 | 17 | -63.0 |
| | PERI | P=0.052 | 3513 | 2659 | NS | 4 | 5 | -1 | 83 | 90 | -7 | 59 | 44 | 25.4 | 53 | 90 | 57 | 90 | 38 | 34 | -10.5 |
| | TH | * | 3471 | 2557 | 0.052 | 2-3 | 2-3 | 0 | 82 | 90 | -8 | 61 | 45 | 26.2 | 53 | 90 | 58 | 90 | 38 | 33 | -13.2 |
| | HAB | NS | 2379 | 1992 | NS | 9 | 11 | -2 | 89 | 90 | -1 | 45 | 36 | 20.0 | 54 | 90 | 58 | 90 | 37 | 33 | -10.8 |
| | HY | NS | 7908 | 6766 | NS | 4 | 4 | 0 | 39 | 34 | 5 | 123 | 105 | 14.6 | 23 | 59 | 21 | 54 | 37 | 34 | -8.1 |
| Brain stem | MB | * | 6125 | 4013 | * | 3 | 4 | -1 | 66 | 83 | -17 | 87 | 56 | 35.6 | 42 | 84 | 53 | 90 | 43 | 38 | -11.6 |
| | PAG | NS | 25740 | 22474 | NS | 8 | 10 | -2 | 43 | 38 | 5 | 350 | 311 | 11.1 | 35 | 75 | 31 | 61 | 41 | 31 | -24.4 |
| | HB | NS | 9291 | 8124 | NS | 2-3 | 2-3 | 0 | 42 | 38 | 4 | 133 | 114 | 14.2 | 28 | 66 | 27 | 61 | 39 | 35 | -10.3 |
| Nuclei / tracts | CP | NS | 1736 | 1167 | NS | 4 | 8 | -4 | 89 | 90 | -1 | 39 | 25 | 35.9 | 54 | 90 | 58 | 90 | 37 | 33 | -10.8 |
| | WM | ** | 6953 | 5445 | ** | 4.5 | 6.5 | -2 | 51 | 53 | -2 | 105 | 80 | 23.8 | 37 | 73 | 38 | 69 | 37 | 32 | -13.5 |
| | CM | NS | 23276 | 19903 | NS | 1 | 1 | 0 | 24 | 24 | 0 | 397 | 339 | 14.6 | 15 | 50 | 16 | 47 | 36 | 32 | -11.1 |
| CSF space | PCS | NS | 15163 | 12498 | NS | 2-3 | 2-3 | 0 | 41 | 31 | 10 | 224 | 187 | 16.5 | 23 | 59 | 20 | 50 | 37 | 31 | -16.2 |
| | 3V | NS | 12228 | 10441 | NS | 5 | 6 | -1 | 45 | 46 | -1 | 180 | 151 | 16.1 | 34 | 76 | 33 | 61 | 43 | 29 | -32.6 |
| | LV | NS | 8288 | 6555 | NS | 6 | 11 | -5 | 90 | 90 | 0 | 157 | 123 | 21.7 | 54 | 90 | 58 | 90 | 37 | 33 | -10.8 |
| Caudal | SSS | * | 5832 | 1430 | ** | 1 | 3 | -2 | 64 | 88 | -24 | 77 | 22 | 71.4 | 38 | 90 | 75 | 90 | 53 | 16 | -69.8 |
| | CB | NS | 7369 | 6151 | NS | 6 | 7 | -1 | 47 | 59 | -12 | 97 | 80 | 17.5 | 29 | 75 | 31 | 67 | 47 | 37 | -21.3 |
| MEDIAN ±SD | | | 6125±1653¶ | 4013±1306¶ | | 5.0±2.2 | 6.5±2.8 | | 63±23 | 81±27 | | 56 | 87 | 24±19 | | | | | 37±7 | 32±7 | |
| KO vs. WT difference (Wilcoxon signed-rank test) | | | WT >KO, p<0.0001 | | | WT <KO, p<0.01 | | | WT ~KO, p=0.2971 | | | WT >KO, p<0.0001 | | | | | | | WT >KO, P<0.0001 | | |

## AQP4 distribution determines regional differences in the parenchymal fluid flow

Increased AQP4 KO brain volume is associated with decreased ventricular volumes, increased slow MR diffusion markers, reduced glymphatic transport, and small region-specific, non-significant differences in the vascular density (*Figure 2F*). Using immunohistochemistry for AQP4, we found heterogenous expression in WT animals, with the highest expression in the hippocampus, thalamus, and habenula (*Figure 2H*; *Yao et al., 2008*; *Hubbard et al., 2015*). Next, we asked which factor reflected the changes in MR-DWI or DCE-MRI derived markers more: vascular density or local AQP4 expression?

The vascular density did not correlate with ADC or IVIM measures in both KO and WT animals (*Figure 2G, Appendix 1—figure 1D*). However, when investigating tracer accumulation dynamics, we found vascular density having a low to moderate correlation with arrival time (Spearman's rho=0.6715, p=0.0201) and moderate to high correlation with time-to-peak in WT animals (rho=0.8411, p=0.0013; *Appendix 1—figure 2C*). In KO animals, only a low range correlation with time-to-peak was identified (rho=0.6084, p=0.0398). Interestingly, in WT mice AQP4 expression did not correlate with changes in the MR tracer dynamics (*Appendix 1—figure 2D*), but had a moderate linear correlation with both ADC (r=0.77, p=0.0092) and D (r=0.74, p=0.0137; *Figure 2I*). Finally, there was no correlation found between the vascular density and AQP4 expression (both Pearson's and Spearman's correlation absolute value <0.52 and p>0.16), or between MR diffusion and tracer dynamics parameters (correlation value <0.5 and p>0.05 for any comparison) except for low correlation between the area under the DCE curve and ADC for both genotypes (WT: r=0.63, p=0.0272; KO: r=0.64, p=0.0239). Thus, high vascular density predicts fast CSF tracer influx consistent with the notion that CSF is pumped in along the perivascular spaces surrounding arteries (*Iliff et al., 2012*; *Iliff et al., 2013*). Also, the correlation between slow MR diffusion, ADC and D, with AQP4 density across 10 regions in WT mice (*Figure 2I*) suggests that AQP4 expression is higher in regions with relatively larger interstitial fluid volume, possibly reflecting that AQP4 is upregulated in response to stagnation of interstitial fluid in wildtype mice.

## Discussion

Using non-invasive high-resolution MR CSF space volumetry and cisternography *in vivo*, we found increased brain volume and decreased CSF pool volume, mainly in the ventricular compartment, in mice genetically lacking the water channel AQP4, alongside increased brain-water content (*Table 4*). Changes in brain water content and CSF pool size were not explained by changes in CSF production or the volume of the larger perivascular CSF spaces. Next, we investigated the brain water mobility in AQP4 KO animals using standard MR-DWI and IVIM-DWI. Measures of fast MR diffusion and vascular density were also unchanged between KO and WT mice, although KO animals exhibited a higher variability in vascular density. Slow diffusion (ADC and D) estimates were increased within the parenchyma of KO animals and so was the cortical interstitial space volume measured using the real-time ionophoresis TMA technique. Finally, we asked whether AQP4 expression or local vascular density correlated to slow diffusion, fast diffusion (IVIM), or with measures of decreased CSF MR tracer influx into the AQP4 KO brains. In WT animals, slow diffusion measures were correlated with AQP4 expression and differential vascular density was nonlinearly correlated to measures of tracer accumulation. AQP4 KO animals had a very low correlation of vascular density to time-to-peak tracer accumulation. These correlations suggest that the vascular network provides a highway for perivascular CSF inflow and thereby drives the initial tracer distribution within the parenchyma. Increased AQP4 expression in regions manifesting high ADC or D in WTs, possible due enlarged interstitial volume, may reflect a compensatory upregulation of AQP4 due to fluid stagnation consistent with the notion that AQPs reduce parenchymal resistance and facilitate the water and solute movement. Consistent with this hypothesis, recent studies report dynamic AQP4 relocalization leading to changes in signaling pathways (*Salman et al., 2022*). Thus, our data overall suggest that the markedly altered brain fluid transport in AQP4 KO mice may result from a reduction in glymphatic fluid export, leading to stagnation of ISF and enlargement of the interstitial space. The interstitial fluid stagnation will in turn reduce CSF influx and give rise to an overall reduction in glymphatic transport.

Why would AQP4 play a role in export of brain interstitial fluid? In general, AQPs increase membrane water permeability and are present in kidney and exocrine organs where fluid transport is

**Table 4.** Descriptive summary of findings presented in the current study.

Bold italic font highlights the regions of the largest differences found between KO and WT animals, by means of 3 MRI and 5 physiological and histological assessment methods applied.

### Magnetic resonance imaging *in vivo*

| | Measurement | General findings in KO compared to WT | Region of largest difference |
|---|---|---|---|
| | 3D cisternography | - 5–10% larger brain volumes<br>- 22–29% smaller CSF space / brain volume ratio | ventricular space, **3rd ventricle** |
| non-invasive | 2D diffusion-weighted imaging | 5–6% higher ADC and D | (difference present for average and in all diffusion directions)<br>- **thalamus**, hindbrain, periaqueductal gray regions, auditory cortex and **hippocampus** |
| | | Higher Fp and Fp × D* only in the **3rd ventricle** | |
| invasive | Dynamic CSF tracer imaging via cisterna magna` | reduced parenchymal tracer influx and evacuation | - influx: cortical ROI, **hippocampus**<br>- efflux: superior sagittal sinus |

### Physiological and histological measurements

| | Measurement | General findings in KO compared to WT | Region of largest difference |
|---|---|---|---|
| | Brain water content | ~6% larger brain water content | |
| | AQP4 expression | (only WT) heterogenous AQP4 expression in the brain | largest expression in the **thalamus, hippocampus,** habenula |
| ex vivo and histology | Vascular density | similar vascular density to WT | trend for larger vascular density in the **thalamus** and olfactory area |
| | CSF production | similar CSF production to WT | |
| in vivo | Real-time ionophoresis TMA | ISF space volume larger | |

driven by relatively small osmotic gradients produced by plasma membrane ion transporters (*Salman et al., 2022*; *Salman et al., 2021*). AQPs facilitate near-isomolar transepithelial fluid transport and AQP deletions have previously been shown to be associated with reduced fluid secretion (*Verkman, 2009*). A recent study showed that AQP4 sharply reduces outflow of interstitial fluid (*Plá et al., 2022*). It has been shown that brain interstitial fluid leaves by multiple pathways including perivenous efflux, along cranial and spinal nerves and also along the ventricular and pial surfaces (*Rasmussen et al., 2022*). The fluid is then absorbed by meningeal and cervical lymphatic vessels for export to the venous system. AQP4 is intensely expressed in glia limitans facing the ependymal layers of the ventricles and also the pial membrane surrounding the brain surfaces. Recent studies showed dynamic and reversible AQP4 channel membrane relocalization for regulation of local water homeostasis (*Salman et al., 2022*) in response to hypothermia and hypotonic treatment in cultured rat (*Kitchen et al., 2015*) and human primary cortical astrocytes (*Salman et al., 2017*) without change in AQP4 mRNA levels, and nonuniformity of these responses among AQP4 subtypes (*Ciappelloni et al., 2019*). Similarly, the vascular endfeet of astrocytes plastered around both arterioles, capillaries and veins in both rodent and human brain express high levels of AQP4 in the membrane facing the vessel wall (*Nielsen et al., 1997*; *Rash et al., 1998*; *Oberheim et al., 2009*; *Zeppenfeld et al., 2017*). In fact, the intensity of AQP4 immunofluorescence signal in endfeet plastered around veins is almost double of those surrounding arteries (*Iliff et al., 2012*). Fibrous astrocytes in white matter tracts are also recognized for their high AQP4 expression (*Lundgaard et al., 2014*). Thus, AQP4 is present at high density at most if not all glymphatic efflux paths, and therefore also in a prime position to mediate outflow of interstitial fluid.

Some of the most critical findings presented here were that the changes in the brain water content, CSF pool size, and interstitial spaces were not due to changes in CSF production in AQP4 KO mice or in perivascular space volume. This is consistent with growing evidence that CSF distribution is dependent on arousal state and circadian timing rather than being dictated by the rate of CSF production (*Hablitz et al., 2020*; *Xie et al., 2013*; *Liu et al., 2020*). Furthermore, vascular density or fast diffusion estimates were not altered by AQP4 deletion, suggesting that the vasculature and blood perfusion remains the same even after genetic deletion of AQP4.

Instead, we show that slow MR diffusion measures are increased mostly due to an enlarged interstitial space. Only slight 5–6% increase in the mean ADC and D might result from superposition of opposing effects of reduced transmembrane permeability (reducing ADC) and increased extracellular space (increasing ADC) as concluded previously using time-dependent diffusion MRI and Latour's model of long-time diffusion behavior (*Pavlin et al., 2017*). Similarly, evaluation of ADC using multi-b-value-multi-diffusion-time DWI provided higher ADC's in healthy hemispheres of mice subjected to contralateral ischemic stroke, reflecting larger interstitial space in KO (*Urushihata et al., 2021*). The enlarged interstitial space in both awake and anesthetized AQP4 KO mice is also consistent with previous reports under anesthesia (*Yao et al., 2008*; *Amiry-Moghaddam et al., 2003b*; *Papadopoulos et al., 2004*). Our findings in the water phantom filled with Sephadex microbeads of similar porosity but different sizes also confirmed increase in both ADC and D resulting from increased free fluid volume (*Figure 2D*).

Our findings support a glymphatic model whereby cerebrospinal fluid is driven by vascular movement into the brain alongside the perivascular space, and AQP4 at the vascular astrocytic endfoot enables ISF and solute movement from the parenchyma. These changes in the micro- and macroscopic CNS fluid compartments could be due, specifically, an increased resistance towards glymphatic fluid efflux caused by lack of AQP4 channels along the perivenous space (*Xie et al., 2013*; *Plá et al., 2022*). Fluid accumulation in the interstitial space would, in turn, increase resistance toward periarterial CSF influx explaining the overall suppression of glymphatic transport.

One group has previously published evidence against the importance of AQP4 in glymphatic fluid transport (*Smith et al., 2017*; *Smith and Verkman, 2018*). This group's finding is contradicted by multiple independently generated datasets using different transgenic lines of mice with deletion of the AQP4 or α-syntrophin (Snta1) genes, different fluorescent and radioisotope-labeled tracers of 4.5–70 kDa size (*Mestre et al., 2018a*; *Hablitz et al., 2020*) or $H_2^{17}O$ at 9.4T MRI (*Zhang et al., 2019*) across a wide age range of 6–24 weeks. A meta-analysis of all published studies demonstrated a significant decrease in CSF tracer transport in AQP4 KO mice compared to wildtype and meta-regression suggested that differences in anesthesia, age, and tracer delivery explained the opposing results.

Since we have discussed the controversy in details (*Mestre et al., 2020*), the only additional note is that observations reported here add additional support to a key role of AQP4 in glymphatic flow.

Glymphatic disruption has been observed in preclinical models of Alzheimer's disease (AD; *Peng et al., 2016*; *Harrison et al., 2020*). In humans, alterations in MR-DWI have been seen in mild cognitive impairment and symptomatic AD (*Kantarci et al., 2001*; *Kantarci et al., 2002*; *Kulkarni et al., 2020*). Here, a linear correlation between vascular density and time-to-peak tracer accumulation across multiple brain regions was found in both KO and WT mice. We also found overall increased slow water diffusion in AQP4 KO mice, which was most pronounced in regions that normally exhibit higher AQP4 expression. These observations suggest that vascular density determines the speed of tracer distribution whereas AQP4 expression determines interstitial fluid exchange. Interestingly, increased slow MR diffusion is also found in AD patients, compared to those with mild cognitive impairment and healthy controls (*Bergamino et al., 2020*). Perhaps known alterations in AQP4 distribution and abundance in AD (*Zeppenfeld et al., 2017*; *Boespflug et al., 2018*; *Simon et al., 2018*) drive increased interstitial fluid stagnation, decreasing glymphatic function past what is expected in natural aging (*Kress et al., 2014*). Extending this speculation, perhaps regional loss of AQP4 may explain subregion-dependent susceptibility to neurodegeneration by driving local interstitial fluid and protein stagnation increasing the risk of aggregation prone proteins. The complex approach we used here, based on cutting-edge non-invasive MR techniques including high-resolution 3D non-contrast cisternography with sophisticated automatic CSF volume estimation, DWI along with IVIM-DWI, standard DCE-MRI, and traditional *ex vivo* histology and physiological measures, may have potential to answer some of these fundamental questions on how cellular pathology and glymphatic dysfunction contributes to proteinopathies.

## Materials and methods
### Animals and experimental setups

Animal approval was received from both the University of Copenhagen Animal Experiment Inspectorate and the University of Rochester Medical Center Committee on Animal Resources. The same AQP4 knockout (*Aqp4(-/-)*, KO) mouse line (*Thrane et al., 2011*), regularly cross-breed with wildtype (WT) mice was bred in both Copenhagen and Rochester, and in total n=97 10–16 weeks old AQP4 KO and WT (*Aqp4(+/+)*) littermates mice on a C57BL/6 background (*Mestre et al., 2018a*) was used (*Table 1A*). All animals were group-housed (up to 5 mice/cage) with ad-libitum access to food and water, temperature (22 ± 2°C), and humidity-controlled (55 ± 10%) environment with a 12/12 hr light/dark cycle. The animals were subdivided randomly into groups, and each group underwent one of the experimental paradigms including three *in vivo* magnetic resonance imaging (MRI) experiments: 3D CSF space volumetry and cisternography, 2D diffusion-weighted imaging (DWI), dynamic contrast-enhanced (DCE) MRI via cisterna magna; or *in vivo* CSF production, and three *ex vivo* measurements: vascular density, AQP4 expression, brain water content. At the end of each *in vivo* experiment, mice were sacrificed via K/X overdose and cervical dislocation.

### MRI

All MRI scanning was performed at 9.4T MRI device (BioSpec 94/30USR, Bruker BioSpin, Ettlingen, Germany) in the head-first prone position with animal's body temperature maintained at 37 ± 1°C with a thermostatically-controlled waterbed and monitored, along with the respiratory rate, by an MR compatible remote monitoring system (SA Instruments, NY, USA).

### Non-contrast MRI volumetry and cisternography

To achieve high-spatial resolution of MRI CSF space volumetry and cisternography a 3D constructive interference in steady-state (CISS) sequence along with a cryogenically cooled Tx/Rx quadrature-resonator (CryoProbe, Bruker BioSpin) and 240 mT/m gradient coil (BGA-12S) were used. During MRI, the animals (5 KO and 6 WT) were anesthetized under Ketamine/Xylazine (*i.p.* K/X: 100/10 mg/kg) and underwent acquisition of two 3D-TrueFISP volumes of opposite phase encoding direction (i.e.: 0° and 180°) (*Table 1B*), for further 3D-CISS image calculation. The complete MRI protocol lasted over an hour so every animal was implanted with a permanent intraperitoneal PE-10 catheter in the abdominal area, connected to a 1 mL syringe filled with K/X solution. The syringe was kept outside

MR during whole imaging, and animals received a single supplementary dose of K/X after the first TrueFISP volume acquired. No difference in age (p=0.697 for KO vs. WT using Mann-Whitney U-test), body weight (p=0.7662), respiration rate during MRI (p>0.99) as well as signal-to-noise ratio (SNR) of computed 3D-CISS images (3.5 ± 0.4 vs 3.7 ± 0.2; p=0.1385) was found between KO and WT animals (*Table 1A*) so no animals were excluded from further analysis.

## DWI and IVIM-DWI

To assess differences in the brain water mobility between the genotypes, echo-planar-imaging (EPI)-based diffusion-weighted imaging (DWI) was performed using a room-temperature volumetric Tx/Rx resonator (in. ø40 mm) and 1500mT/m gradient coil (BFG6S, Bruker). The animals (6 KO and 6 WT) were anesthetized with K/X (*i.p.* 100/10 mg/kg) and underwent DWI with 17 b-values measured in 3 orthogonal directions of diffusion encoding gradients (*Table 1B*). To reduce the effect of respiratory motion (*Federau et al., 2013*), all DWI images were acquired with respiratory-gating in exhale, assisted by the remote monitoring system (see above). To minimize the influence of deep anesthesia and long scanning time on the measurements, the imaging protocol encompassed solely DWI lasting <40 min so no supplementary K/X was required. No difference in age (p>0.99 for KO vs. WT; Mann-Whitney U-test), body weight (p=0.935), respiration rate during MRI (p=0.632) was found between KO and WT so no animals were excluded from further analysis (*Table 1A*).

It is worth mentioning, that there is an ongoing debate on efficacy of IVIM modelling in reflecting phenomena in microvascular network or related to tissue microarchitecture (*Fournet et al., 2017*; *Meeus et al., 2017*; *Paschoal et al., 2018*; *Schneider et al., 2019*; *Niendorf et al., 1996*). We have aimed to provide an optimal setup for DWI (*Liao et al., 2021*; *Lemke et al., 2011*) by measuring MR diffusion signal using 30ms echo time (TE) and 17 b-values with increased averaging for b-values≥1000 s$^2$/mm (see *Table 1B*). Based on or preliminary assessment (unpublished) application of higher than minimal available (here minimal ~22ms) TE would reduce the influence of ghosting and perfusion-related artefact, and higher averging would reduce the influence of possible Rician noise at high b-values. Although IVIM estimates were reported to depend on TE (*Führes et al., 2022*; *Bisdas and Klose, 2015*), mostly for perfusion fraction, our evaluation focused predominantly on the slow diffusion component. Furthermore, by sampling signal up to 3000 s$^2$/mm b-value, which may lead to presented slight lower ADC values, we aimed for depicting dominant signal from extracellular space at high b-values (*Le Bihan, 2019*; *Cihangiroglu et al., 2009*; *Clark and Le Bihan, 2000*; *Niendorf et al., 1996*). However indicatively useful, higher order models and the models focused on separating hindered MR diffusion signal according to assumption on microarchitecture (*Latour et al., 1994*; *Palombo et al., 2020*; *Burcaw et al., 2015*; *Kaden et al., 2016*; *Wu and Zhang, 2019*; *Olesen et al., 2022*; *Pfeuffer et al., 1998*) or assessment of diffusion signal distribution (*Roth et al., 2008*; *Benjamini and Basser, 2019*; *Slator et al., 2021*; *Ronen et al., 2006*) are beyond presented general evaluation.

## CM cannulation and DCE-MRI

For DCE-MRI, cisterna magna (CM) cannulation based on the previous studies (*Xavier et al., 2018*) was performed in mice (5 KO and 6 WT; *Table 1A*) anesthetized with Ketamine/Dexmedetomidine (*i.p.* K/Dex: 75/1 mg/kg). After exposing CM, a 30 G copper cannula (out. ø0.32 mm; Nippon Tokushukan, Mfg, Tokyo, Japan), attached to a PE-10 tube filled with aCSF, was inserted into CM and fixed in position with a drop of cyanoacrylate glue followed by a drop of glue accelerator. The incision site and the skull were then covered by a mixture of cyanoacrylate glue and dental cement. Subsequently, the exposed PE-10 tubing was attached to a cannula filled with contrast agent (gadobutrol, 20 mM; Gadovist, Bayer Pharma AG, Leverkusen, Germany) and the animals were moved to MR scanner. The filled cannula was then connected to a syringe in a microinfusion pump. Head movements during the scanning were minimized by fixing the animals head in an MR-compatible stereotactic holder with ear bars, and the animals were put into MR with the head centered under the CryoProbe. The scanning protocol based on the previously described (*Stanton et al., 2021*) and whole mouse brain pre- and post-contrast $T_1$-weighted DCE-MRI was acquired with 1 min temporal and 100 μm isotropic spatial resolutions using 3D-FISP sequence (*Table 1B*). DCE-MRI continued over 90 measurements (90 min), and $T_1$-enhancing contrast agent was infused into the CM (gadobutrol, 1 μL/min for 10 min) after the first three baseline scans (i.e., after 3 min).

## Microbeads phantom *ex vivo*

To verify whether ADC and D may reflect the volume of interstitial fluid (ISF) space in the brain parenchyma, we performed an additional DWI measurements by mimicking increasing ISF space volume in 3 water phantoms filled with Sephadex G-25 microbeads (Sephadex G-25; Sigma-Aldrich, St. Louis, MO, USA) of coarse (100–300 µm), medium (50–150 µm), fine particle size (20–80 µm of wet particle size). All microbeads possess the same porosity <5 kD so a similar exchange rate between stored and free water pools is expected. Each phantom was of the same in-house design (in. volume ~0.5 ml.), formed from of thick plexiglas tube (~25 mm in. long, out./in. ø10/5 mm) with a thread plug at both sides. The thread plug was made from the same tube, with a standard bonded polyester microfilter placed inside (*Appendix 1—figure 1E*) to prevent evacuation of the microbeads. For DWI, microbeads for each phantom were placed initially in a distilled water for 24 hr, to achieve their maximal size. Afterwards, a 2 ml syringe filled with distilled water solution of 0.001 mM/ml gadobutrol was attached to the thread plug on one side of the phantom and the phantom was filled with the microbeads. Gadobutrol was used to obtain optimal MR signal shortening for the echo time used in the same EPI sequence as employed *in vivo* (*Table 1B*). Any possible air bubbles remaining between the microbeads were carefully removed using a 1 µl inoculating loop. Subsequently, the other side of the phantom was closed with the thread plug, the phantom was flushed with ~1.8 ml gadobutrol solution in the attached syringe, and the solitary plug was sealed with a rubber syringe cap and parafilm. The syringe with a residual of gadobutrol solution at the other end was left to support pressure equalization inside the phantom.

DWI was performed 6 times in each phantom, with a central slice imaged in 1/3 phantom's distal portion from the syringe. To verify the diffusion values in a free water environment, DWI was performed 4 times in the same phantom filled solely with gadobutrol solution.

## Free fluid volume estimation in the phantom

To confirm that the free fluid volume surrounding the microbeads increases with increasing particles size, additional measurements were performed employing MR $T_1$ mapping and contrast-enhanced micro computed-tomography (µCT).

For $T_1$ mapping, all phantoms were prepared *de novo* and scanned jointly using spin-echo sequence (2D-RARE; *Table 1B*) with variable repetition times. For each phantom, dry microbeads were put for 24 hr directly into the phantom lumen filled with 0.001 mM/ml gadobutrol solution. Before MRI, any possible air bubbles were removed, and the phantom was sealed (as above). Microbeads of different size were expected to differently infiltrate surrounding water, and partially gadobutrol. Thus, the relative concentration of gadobutrol was expected to be altered compared to original solution, and shorter $T_1$ relaxations were expected to be observed for phantoms with decreasing microbeads size. For reference, $T_1$ mapping was performed in the phantom filled solely with gadobutrol solution. To correct $T_1$ maps for $B_1$-filed inhomogeneities using double-angle method (*Insko and Bolinger, 1993*; *Cunningham et al., 2006*), all phantoms were scanned with the same spin-echo sequence with a maximal repetition time used for $T_1$ mapping, and using two different flip angles (*Table 1B*).

For free fluid space estimation using µCT, all phantoms previously scanned for DWI were unsealed and flushed with a distilled water solution via 5 ml syringe to remove gadobutrol. Subsequently, distilled water was replaced with a 1:1 dilute of non-ionic iodine Omnipaque 350 contrast agent (Iohexol, 350 mg iodine/ml; GE Healthcare AS) in normal saline via 3 ml syringe. Each phantom was scanned using Vector4uCT system (MILabs, Utrecht, Netherlands) using the scan parameters of 15 µm isometric resolution, 50 kVp, 0.24 mA (75ms exposure), 360 degrees rotation, 0.2 degree rotation step, 2 frames for averaging, 0.5 mm thick beam aluminum filter, Hann filter with cone-beam reconstruction.

To verify the relation between MR diffusion values and the $T_1$ relaxation times as well as free fluid estimates using µCT, DWI was performed 4 times in each phantom (as above).

## AQP4 channel staining *ex vivo*

C57Bl/6 brain sections collected previously (*Hablitz et al., 2020*, ) were newly reimaged and an entirely new analysis done for current experiment. Originally, mice (4 WT; *Table 1A*) were cardiac perfused with AlexaFluor 488 conjugated wheat germ agglutinin (Thermofisher Scientific) at 15 µg/mL in 20 mL of phosphate buffered saline (PBS), and then perfused with 20 mL PBS with 4% paraformaldehyde (PFA). Subsequently, the brains were extracted and were immersed in 4% PFA overnight.

This step labels the vasculature and fixes the brain tissue. Brains were sectioned coronally at 100 µm using a vibratome (Leica VT1200s) and equivalent sections from +1.2 to –1.8 bregma were stained for AQP4. Floating sections were permeabilized with 0.1% Triton-X-100 in PBS, blocked with 7% normal donkey serum (Jackson Immunoresearch) in PBS with 0.03% Triton-X-100, and then incubated with Anti-AQP4 primary antibody (AB3594, Millipore; 1:1000 dilution) overnight. The sections were afterwards washed three times with PBS and incubated with Alexa 594 fluorophore-linked donkey anti-rabbit secondary antibody (A21207, Invitrogen; 1:500 dilution) and DAPI (D1306, Invitrogen; 1:2000 dilution). Stained sections were mounted with Fluoromount G (Thermofisher Scientific).

## Vascular density measurements *ex vivo*

Vascular density was measured in mice brains (6 AQP4 KO and 6 C57Bl6; *Table 1A*) prepared in few steps. All mice were cardiac perfused with AlexaFluor 488 conjugated wheat germ agglutinin at 15 ug/mL in 20 mL of PBS, and then perfused with 20 mL 4% PFA. Afterwards, the brains were extracted and immersed in 4% PFA overnight. 100 µm coronal sections were taken using a vibratome (Leica 1200 S). Anterior brain sections at +1.2 mm bregma, and posterior brain sections –1.8 mm bregma were mounted in Prolong Gold media with DAPI (Invitrogen). Equivalent sections were used for all biological replicates.

## CSF production measurements *in vivo*

The measurements were performed in mice (5 KO and 6 WT; *Table 1A*) as described previously (*Liu et al., 2020*). Mice were anesthetized with K/X (100 mg/kg / 20 mg/mL i.p.) and 2% isoflurane, and placed in a stereotactic frame. Their scalp was shaved, and the skin was cleaned with a chlorhexidine swab followed by an alcohol wipe to remove the chlorhexidine. An iodine solution was applied and left to dry, and the scalp was opened and the skin retracted. The exposed skull was irrigated with sterile saline and cleaned by applying sterile cotton swabs, and a sterilized stainless-steel light-weight head plate (0.9×19×12 mm dimension), equipped with a round hole of at the center (in. ø9.0 mm), was attached to the mouse skull using a mixture of dental cement with cyano-acrylate glue (*Sweeney et al., 2019*). Pre-operatively as well as for three days post-surgery the mice received Banamine (1.1 mg/kg) subcutaneously as an analgesic. The mice were trained to tolerate positioning in the head plate stand (Cat# MAG-1, Narishige International USA Inc), as well as a restraint tube in three daily training sessions, each lasting 30 min for 3 days post-surgery.

For measuring CSF production during wakefulness, the mice were anesthetized with 2% isoflurane during cannula implantation. A cannula (30 G needle) attached to artificial CSF filled PE-10 tubing was implanted into the right lateral ventricle through a small burr hole (AP = –0.1 mm, ML = 0.85 mm, DV = –2.00 mm from the postion of bregma). The cannula was fixed to the skull with dental cement and the opposite end of the PE tubing was sealed by high-temperature cautery. Once the cannula was in place, the neck was flexed 90 degrees and the headplate was attached to the head stand. Then a separate cannula was inserted into the CM and advanced into the 4th ventricle. One microliter of mineral oil was infused into the 4th ventricle to block the exit of CSF into the subarachnoid space. All incisions were infiltrated with 0.25% bupivacaine topical anesthetic to prevent the animal from experiencing post-surgical pain. The measurements were collected while the animal rested in an open cylinder restraint tube (9 cm length, in. ø3.5 cm), to which the animals were accustomed during training. Anesthesia was discontinued and CSF production was measured in head-fixed mice in 10 min intervals for 65 min, after a 30-min recovery period.

## Tetramethylammonium microiontophoresis for interstitial fluid space volume estimation

Real-time iontophoresis with tetramethylammonium (TMA) was performed in mice (8 KO and 20 WT, body weight not recorded) as adapted from the previous studies (*Nicholson, 1993*; *Xie et al., 2013*). The single barrel iontophoresis microelectrode (tip out. ø2–3 µm) contained 150 mM tetramethylammonium (TMA)-chloride and 10 µM Alexa 488. A series of currents of 20 nA, 40 nA and 80 nA were applied by a dual channel microelectrode pre-amplifier. For measurements of TMA, microelectrodes (out. ø2–3 µm) were fabricated from double-barreled theta-glass using a tetraphenylborate-based ion exchanger. The TMA barrel was backfilled with 150 mM TMA chloride, while the reference barrel contained 150 mM NaCl and 10 µM Alexa 568. All recordings were obtained by inserting the two

electrodes to a depth of 150 μm below the cortical surface. Recording electrodes were inserted 2.5 mm lateral and 2 mm posterior to bregma. The electrode tips were imaged after insertion using 2-photon excitation to determine the exact distance between the electrodes (typically ~150 μm). The TMA signal was calculated by subtracting the voltage measured by the reference barrel from the voltage measured by the ion-detecting barrel using a dual-channel microelectrode pre-amplifier. The Nikolsky equation was used for calibration of the TMA electrodes based on measurements obtained in electrodes containing 0.5, 1, 2, 4, and 8 mM TMA-chloride in 150 mM NaCl. The TMA measurements were acquired relative to similar recordings obtained in 0.3% agarose prepared from a solution containing 0.5 mM TMA and 150 mM NaCl. A custom-made software in Matlab (v. R2019a, The Mathworks, Inc, Natick, MA.), 'Walter', developed by C. Nicholson was used to calculate α and $\lambda$ values (*Nicholson, 1993*; *Xie et al., 2013*).

## Brain water content *ex vivo*
Anesthetized animals (3 KO and 5 WT, gender and body weight not recorded; *Table 1A*) were decapitated, the whole brains were taken out and weighed immediately ($W_{wet}$). Brain tissue was dried at 65 °C for 48 h until it reached a constant weight, and brain were re-weighed ($W_{dry}$). The ratio between the difference of $W_{wet}$ - $W_{dry}$ and $W_{dry}$ was considered reflecting the brain water content (ml $H_2O$/g dry weight).

## Data processing and statistical analysis
### MRI
All acquired images were visually checked and no presence of significant artefacts influencing morphological and functional assessment was found. Further processing pipelines were applied, and included motion-correction, spatial co-registration, and automatic or semi-automatic pre- and final postprocessing.

All described statistical analyses were performed in GraphPad Prism 8 (GraphPad Software) and Matlab. The results coming from statistical comparisons were considered significant for $p < 0.05$ after post-hoc correction, when applicable.

### 3D-CISS volumetry and cisternography
All 3D-CISS volumes were calculated in few steps using in-house pre-processing pipeline (*Appendix 1—figure 3A*). For each animal, every 3D-TrueFISP volume acquired with two repetitions was motion-corrected (10 times or until no further improvement) and averaged. Subsequently, the second averaged 3D-TrueFISP volume (180° encoding direction) was subjected to rigid-body registration (6 df.) to the first volume (0° encoding direction). Both motion-correction and registration were performed in AFNI (*Oakes et al., 2005*), and aimed to reduce the influence of random motion on the computed 3D-CISS image. Finally, 3D-CISS image was computed as a maximum intensity projection from 2 co-registered 3D-TrueFISP volumes, resulting in an image of almost completely removed banding artifacts. Afterwards, every computed 3D-CISS volume underwent semi-automatic brain parenchyma image extraction using the 'Segment 3D' tool in ITK-SNAP (*Yushkevich et al., 2006*), to remove the regions outside the brain parenchyma image from further analysis. Brain parenchyma was considered as the brain tissue volume surrounded with dark regions of skull image and including intracerebral vessels. To correct for intensity inhomogeneities coming from the $B_0$ field and the surface profile of the CryoProbe ($B_1$), the extracted brain parenchyma image underwent bias field correction using FSL (*Zhang et al., 2001*) (0.5 sigma, 20 mm FWHM, 4 iterations). At each step of pre-processing, the results were visually checked and confirmed for correctness.

### Automatic CSF space segmentation
For a single, bias-corrected 3D-CISS volume the ventricular and perivascular CSF spaces were separated from the brain parenchyma image in 3 dimensions using an in-house fully automatic adaptive algorithm in Matlab (*Gomolka et al., 2021*), in four steps (*Appendix 1—figure 3B*). First, high-intensity regions, as branches of the optic nerve's residual after the semi-automatic brain extraction and not adjacent to the parenchyma in all image slices, were excluded using a bounding box enclosing the brain (*Appendix 1—figure 3B* 'Bounding box and adaptive thresholding'). The bounding box was

automatically computed based on a maximization of the voxels intensity variance slice-wise, separately in three orthogonal planes. The parenchyma volume surrounded by the bounding box was enclosed and the solitary regions were removed based on their geometrical properties calculated slice-wise in the sagittal plane: eccentricity≥0.5, roudness≥0.5, perimeter<0.005% of the brain parenchyma voxels count. The resulting brain image mask was geometrically dilated with a disk kernel of 11 pixels diameter, to enclose potentially removed or non-continuous parenchymal regions. The non-continuity appeared in case of residuals from banding artifacts at the borders of the skull and the ethmoidal bone. Subsequently, brain parenchyma volume was updated according to the resulting mask, for further automatic segmentation of the CSF space.

Second, initial CSF space separation was performed by means of an adaptive intensity threshold and calculation of cumulative distribution of voxels intensities>0 from the separated and bias-corrected brain volume. As the overall distribution of the brain intensities differed slightly between the 3D-CISS images due to their SNR, a threshold-correcting factor (denoted $f_c$) was calculated by means of the formula:

$$f_c = \frac{\sigma_d}{\mu_d + \sigma_d} \tag{1}$$

where $\mu_d$ is a mean and $\sigma_d$ is a standard deviation (SD) of aggregated brain parenchyma voxels intensity distribution. Separation of the CSF space was performed assuming that CSF intensities reflect those ≥95th percentile of the aggregated intensities distribution. Hence, the intensity of each voxel was rescaled according to the formula:

$$I_r = \frac{(I - 1.33 \times \sigma_d)}{\sigma_d}, \tag{2}$$

where $I$ and $I_r$ are the original and the rescaled voxels intensities. The rescaled voxels intensities were encoded using a floating point precision in range between $-1.33 \times \sigma_d$ and a distribution peak close to the image SNR calculated as $\mu_d / \sigma_d$ (i.e. maximum rescaled intensity ~10 with the mean value varying between 3 and 5 among all analyzed images). Third, all the rescaled voxels possessing negative intensity (i.e. brain parenchyma) were assigned to 0, and a new aggregated distribution of the rescaled voxels of >0 intensity was computed. Subsequently, all voxels intensities ≤95.5th percentile of the new distribution were assigned to 0 to keep only the high intensity CSF seed regions. For the images of SNR >4 (i.e. lower contribution of Rician noise), the new distribution threshold was set of ≤95.5 +$f_c$. The correction factor $f_c$ accounted for subtle intensity changes and did not result in the threshold exceeding the 97th percentile. A mask image of the initially separated CSF space was computed by assigning all remaining nonzero voxels to 1, for further application of a region-growing algorithm.

The final segmentation was performed using a 2D region-growing algorithm applied slice-wise, consecutively in sagittal, axial and coronal planes (*Appendix 1—figure 3B*- 'in Sagittal plane', 'in Axial plane', 'in Coronal plane'). In each slice, the algorithm reconsidered the voxels at the boundary of the CSF space in horizontal, vertical and diagonal directions separately. The algorithm based on extension of the method for contrast calculation applied to study properties of hemorrhagic and ischemic regions in clinical CT images (*Nowinski et al., 2014*; *Gomolka et al., 2017*). The calculations were performed considering the initially separated CSF space mask and the original 3D-CISS image, in two steps: (1) CSF mask dilation; (2) CSF mask erosion.

## CSF mask dilation

To assure that only voxels belonging to the CSF space and not affected by the partial volume from the surrounding parenchyma were included into calculation, the contrast was computed for the initially separated CSF boundary voxels reflecting the intensities ≥97.5th - $f_c$ percentile (i.e., $\mu_d + 2 \times \sigma_d$) of the aggregated intensities distribution from not intensity-rescaled brain parenchyma image. The boundary contrast was obtained as a ratio of an absolute difference between considered boundary voxel intensity and the mean intensity of the $n$ consecutive voxels to the left/top/left-diagonal to the sum of the boundary voxel intensity and the mean intensity of the $n$ consecutive voxels to the right/bottom/right-diagonal in the original CISS image (absolute relative CSF/brain parenchyma contrast in horizontal/vertical/diagonal directions, respectively). The contrast was calculated for $n$ from 1 to 4

in the sagittal and from 1 to 3 voxels in the axial and coronal planes. The voxels at $n^{th}$ distance from the boundary were included into the updated CSF mask if their absolute relative contrast was <2%.

## CSF space erosion

The boundary of the updated CSF mask was recalculated, and the voxels were reconsidered using the same method for the contrast calculation (see above). Herein, however, the contrasts were calculated for $n$ from 1 to 2 in the sagittal and from 1 to 3 in the axial and coronal planes, to avoid removing small regions belonging to the perivascular space around the main cerebral arteries. The voxels at $n^{th}$ distance were removed from the updated CSF mask if their intensity in the original CISS image was $\leq 95.5^{th}$ - $f_c$ percentile (for sagittal, and $<95.5^{th}$ - $f_c$ for axial and coronal planes) of the aggregated intensity distribution, and the absolute relative contrast to the respective boundary voxel was >2.5% (for sagittal, and $\geq 2.5\%$ for the other planes). To further remove remaining false-positively segmented single voxels and to enclose wrongly opened larger regions the final CSF mask was subjected to 3D median filtration with a 3×3 voxels kernel (**Appendix 1—figure 3B** 'Filtration and labeling'). Final CSF volume was visually assessed for correctness by RSG and YM, by overlaying with its parental 3D-CISS volume in ITK-SNAP.

## CSF compartments labeling and statistical comparison

The delineated CSF space was separated manually from the final CSF mask in ITK-SNAP into seven compartments (**Appendix 1—figure 3B –** 'Filtration and labeling'), for further statistical comparison: lateral ventricles; third ventricle; fourth ventricle; basilar artery; basal perivascular space at the skull base surrounding the Circle of Willis; parietal perivascular spaces and cisterns (ventrally from the position of posterior cerebral artery, via space neighboring the transverse sinuses and dorsally to the junction of the superior sagittal sinus and transverse sinuses); remaining perivascular space within the olfactory area, surrounding anterior cerebral and frontopolaris arteries, middle cerebral arteries branches, and posterior cisterns including pontine and cisterna magna. For supplementary comparison, the segmented lateral, third and fourth ventricular spaces were considered jointly as the ventricular space, and the basilar, basal and the remaining anterior/posterior CSF spaces were considered jointly as the whole perivascular space. Number of voxels was counted, and the volume of each segment was calculated by multiplying the voxels count by the voxel dimension from the original 3D-CISS image, for subsequent statistical comparison.

To compensate for the brain capsule volume differences and provide a reliable measure of the brain's CSF space volume between animals, a ratio of the CSF to the brain volume (intracranial volume) was calculated for each delineated CSF segment as:

$$Ratio_{CSF\,space} = \frac{CSF_{compartment\,volume}}{Brain_{volume} - CSF_{whole\,segmented\,volume}}. \tag{3}$$

The ratios obtained for each of the CSF compartments, as well as the segmented brain volumes were compared between KO and WT animals using nonparametric Mann-Whitney U-test.

## DWI and IVIM-DWI

### Preprocessing

For every animal, all DWI volumes acquired were subjected to motion-correction in AFNI to avoid influence of random and subtle frame-to-frame image displacements. The motion-correction was performed (4 times or until no further improvement) in reference to the first image acquired using the first b-value (*b0*), and the results were visually confirmed for correctness by RSG. Images acquired with different number of repetitions were averaged according to b-value and diffusion encoding direction, for subsequent calculation of diffusion measures in all 3 directions. Separately, an averaging took place considering only b-values, to calculate an average diffusion images. To normalize the image intensities and reduce influence of nonstationary noise, volume-wise normalization of voxels intensities was performed in all images from different b-values using a mean value of a background signal defined in a half-ball VOI including 704 voxels outside the visible tissue image. The VOI was manually set in an averaged *b0* image and applied to every averaged volume from each b-value for the background signal depiction. The intensity normalization was performed according to formula:

$$DWI_{bval\,norm} = DWI_{bval} - (\mu_{bval\,noise} - 6 \times \sigma_{bval\,noise}), \tag{4}$$

where $DWI_{bval}$ and $DWI_{bval\,norm}$ are the original and normalized voxels intensities, and $\mu_{bval\,noise}$ and $\sigma_{bval\,noise}$ are respectively the mean and SD of the background signal at b-value. To reduce the influence of distortion artefacts, all DWI images were subjected to slice-wise spatial smoothing using a 0.5 sigma [3×3] Gaussian filter. To automatically exclude the voxels outside the brain regions in subsequent diffusion curve fitting, a brain mask image was calculated from $b0$ image and applied to all images from different b-values. The mask image excluded all the voxels below the mean + 0.5 × SD of the $b0$ image intensity. As initial threshold excluded voxels ventrally from the brain image, the mask was subjected to morphological dilation with a diamond shape kernel of 2 pixels diameter, and subsequent image filling to include sole obsolete voxels inside the mask.

## Diffusion-curves fitting

All calculations were performed in Matlab using in-house computational pipelines and curve-fitting implementations based on the least squares method. For a standard DWI model, monoexponential curve-fitting was performed voxel-wise using all 17 b-values. The ADC and estimated $S_0$ images were calculated voxel-wise based on the formula:

$$S_{bval} = S_0 \times e^{(-bval \times ADC)}. \tag{5}$$

For IVIM model, a two-step algorithm was used as considered providing more robust and reliable results compared to standard biexponential curve-fitting (*Lee et al., 2016*). In the first step, a perfusion fraction ($F_p$) was estimated as a voxel-wise ratio of the difference between $b0$ and approximated $S_0$ to $b0$ image. Voxels intensities in the $S_0$ image were estimated from the linear regression approximation considering the logarithm intensities of the volumes from the IVIM threshold to the highest b-value. The threshold b-value used was closest to 250 $s^2/mm$ (here ~238 $s^2/mm$). The slow diffusion D was estimated as the slope coefficient of the linear regression function. Subsequently, for the voxels where a positive and nonzero $F_p$ value was calculated (presence of fast diffusion) a biexponential curve-fitting was performed using the calculated $F_p$ and D:

$$S_{bval} = S_0 \times \left( F_p \times e^{(-bval \times D^*)} + (1 - F_p) \times e^{(-bval \times D)} \right), \tag{6}$$

where D is the slow ('pure') molecular water diffusion, D* is the fast diffusion (pseudodiffusion).

## Regions of interest definition and statistical comparison

For further statistical comparison, the mean and SD of ADC, $F_p$, D and D* signal intensities were calculated within 21 regions of interest (ROI), for every animal brain image separately. The regions were chosen based on reported high Aqp4 expression (*Hsu et al., 2011*; *Hubbard et al., 2015*) and were drawn manually using ITK-SNAP in estimated $S_0$ image, and overlaid on the calculated diffusion parameters maps. The defined ROIs included from 48 to 250 voxels depending on the anatomical structure and excluding neighboring distortion artefact, and covered the position of cortical (olfactory area - OLF; cingulate/retrosplenial cortex - CA/RSP; visual - VIS; somatosensory - SS; auditory - AUD; hippocampus - HIP; perirhinal - PERI), brain stem (thalamus - TH; habenula - HAB; hypothalamus - HY; midbrain - MB; periaqueductal gray - PAG; hindbrain - HB), cerebral nuclei and tracts (caudate putamen - CP; white matter including striatal regions - WM); CSF space (lateral ventricles - LV; third ventricle - 3 V; fourth ventricle - 4 V; pericisternal space - PCS; perivascular space within the Circle of Willis - CoW) and the cerebellar ROI (cerebellum - CB). Application of automatic segmentation template was avoided due to variable influence of distortion and ghosting artefacts at large span of 17 b-values measured, resulting in erroneous template registration and delineation of signal from specific anatomical structures. The regions were set manually by RSG in reference to the Allen Brain Atlas, and verified by YM and MN.

Due to presence of distortion and ghosting artefacts resulting in possible inhomogeneous noise properties of EPI images acquired, voxel intensities within delineated ROIs were considered coming from independent and non-uniform signal distributions (i.e., comparing ventral vs. dorsal ROI). Therefore, mean ADC, D, $F_p$, D* and $F_p$ × D* were calculated excluding <0 and >99 percentiles of their intensity distribution within each ROI (see calculated values in the supplementary Excel file), and were

compared ROI-wise between KO and WT animals using a nonparametric Mann-Whitney U-test. To confirm reliability of the slow diffusion measures, Pearson's linear correlation was calculated between the mean ADC and D from all animals.

## DCE-MRI
### Preprocessing

Time-series of FISP volumes acquired were motion-corrected and spatially normalized with ANTs (*Avants et al., 2014*) and co-registered to the respective baseline image, subject-wise. For every animal, the co-registration process was repeated twice using rigid-body (6 df.) and twice using an affine transformation (12 df.) to assure accuracy. Subsequently, the intensities from the first volume acquired after the dummy scan (i.e., of unstable steady-state MR signal) were subtracted voxel-wise from succeeding volumes to reveal only the regions enhanced with the contrast agent. To normalize the CSF signal in each time-series their voxel intensities were subjected to Gaussian normalization using the first 3D-FISP volume. The resulting images were smoothed with a 3×3×3 voxels kernel of [0.2, 1, 0.2] weights along each axis, to reduce the influence of possible artifacts after the automatic registration and subtracting the baseline volume.

### Volumes of interest definition and statistical comparison

For further statistical comparison, DCE 3D-FISP signal was derived from 21 ROI reflecting those from DWI analysis (see above), in each animal individually. All ROI were defined in a 3D template volume, based on Reference Space Model from Allen Brain Atlas, of the same spatial resolution as 3D-FISP volumes acquired. The reference volume was automatically registered in AFNI using affine registration (6 df.) to the baseline DCE volume. Mean DCE signal was depicted from every 3D-FISP volume in every animal ROI-wise (90 signal intensity values / ROI), for further statistical comparison. The first volume following the dummy scan was rejected from further consideration as it had previously been used for the signal normalization. To compare ROI-wise DCE signals between KO and WT animals, a repeated measurements Two-way ANOVA along with post-hoc Bonferroni's correction was used in search for differences between the signal changes at specific time points and compared to the baseline signal before the gadolinium infusion. Further calculation of DCE-derived scores of CSF tracer arrival time, relative time-to-peak, peak intensity, and duration of significant from baseline interstitial tracer accumulation was performed based on the rank-sum scores provided by Dunn's multiple comparison from nonparametric Friedman's one-way ANOVA, performed separately in KO and WT groups. Arrival time was considered as a difference in time between the baseline time point, and time point where the negative rank-sum score was at least three times that from the baseline. Time-to-peak was calculated as a time difference between the start of the gadobutrol infusion and the subsequent time point at which the maximum negative rank-sum score was calculated (i.e. corresponding to the peak CSF signal intensity within the group). The peak intensity was considered as a relative signal increase occurring at the time-to-peak. Duration of significant from baseline interstitial tracer accumulation was obtained as a time difference between the last and the first time points, where DCE signal increase was significantly different from baseline. Area under the DCE curve was calculated for each subject as a sum of DCE signal amplitudes over all the measured time points. DCE scores derived for KO and WT groups were compared ROI-wise using nonparametric two-tailed Wilcoxon signed-rank test.

## Microbeads MR phantom *ex vivo*

DWI and IVIM measures were computed using the same processing pipeline as for *in vivo* imaging. To avoid influence of artefacts due to field inhomogeneity or at the border of the phantom lumen, all diffusion values were depicted in a 1.5 mm diameter circular region around the center of the phantom image, in all slices. Mean diffusion values from all phantoms were compared using nonparametric Kruskal-Wallis one-way ANOVA with Dunn's post-hoc.

### $T_1$ mapping

Calculation of $T_1$ maps was performed using an in-house algorithm in Matlab. The voxel intensities from spin-echo images of the phantoms were approximated over the increasing repetition times, by inversely solving the equation with the least squares method:

$$S_{TR} = S_0 \times \left(1 - e^{\left(\frac{TR}{T1}\right)}\right) + \varepsilon, \tag{7}$$

where TR is a repetition time, $S_0$ is the estimate of the proton density, $S_{TR}$ is the intensity of the considered voxel from the corresponding slice, and ε is an estimator of a random error. To reduce influence of transmit inhomogeneity on $T_1$ estimations, $B_1$-correction was applied using dual-angle method based on relationship between an effective and nominal flip angles (*Cunningham et al., 2006*) and by applying voxel-wise a correction factor dependent on the intensities from two spin-echo volumes acquired with 45° and 90° flip angles.

## Free water volume fraction

Evaluation of the free water volume fraction inside the phantoms with microbeads was performed based on percentage of the contrast-filled volume, quantified in 8 bits gray scale axial images. First, the central portion of the phantom volume, reflecting that from DWI analysis, was manually separated in ImageJ (v. 1.53 j, NIH, USA). The free fluid volume regions (filled with Omnipaque 350 solution) were of clearly higher Hounsfield Units (HU) intensities compared to those from the microbeads. Thus, the voxels belonging to the free fluid were empirically verified occupying >75th percentile of the cumulative HU distribution in all the separated volume images, from all phantoms. Therefore, the voxels belonging to the free fluid were counted three times, considering only voxels above 75th, 80th, and 85th percentile of HU distribution. For each phantom, the free fluid volume fraction was estimated as a mean voxel count from all thresholds, divided by the voxels count in the separated volume image. As objective separation of the fluid space was impossible in the phantom with the fine microbeads and lower HU thresholds would consider voxels affected by partial volume from the microbeads, proposed free fluid volume fraction estimation was found sufficient to correct for the changes in shape of voxels HU distribution from different phantoms.

## AQP4 staining and vascular density image analysis

Brain sections were imaged using a conventional fluorescence macroscope (Stereo Investigator with objective UplanXApo 10x /numerical aperture 0.40, ∞/compatible cover glass thickness 0.17 mm/ field number 26.5 mm, no immersion liquid (air); Olympus) and subsequently analyzed using ImageJ. Multiple FOVs were acquired in a 1360 × 1024 px / 1392.44 ×1048.43 μm frames (1.048 μm²/pixel), and subsequently aligned and stitched together to provide image of entire brain section. Both AQP4 and vascular staining were assessed in a complimentary anterior and posterior entire brain sections (cf. AQP4 channel staining *ex vivo*). To minimize the reader-associated bias, all image analyses were performed by the investigator (MG) blinded to the image content and animal genotype, and in randomized images.

### AQP4 staining

For each brain hemisphere in each section, all ROIs were manually drawn according to the Allen Brain Atlas and using auto fluorescence from the green channel following visible anatomical landmarks and confirming AQP4 expression visibility. Each subregion measured from 0.03 to 1.69 mm². Therefore, a universal threshold was applied manually to all images as a preferred method for reduction of the background signal influence. Subsequently, a mean area fraction covered by AQP4 in both hemispheres/sections was measured for each ROI. In total, mean AQP4 expression was calculated for 11 ROIs (retrosplenial cortex - RSP, visual - VIS; somatosensory - SS; auditory - AUD; hippocampus - HIP; perirhinal - PERI; thalamus - TH; habenula - HAB; hypothalamus - HY; pericisternal - PCS; white matter - WM) in 4 WT mice. A nonparametric Kruskal-Wallis one-way ANOVA with Dunn's post-hoc was employed to compare the mean AQP4 channel expressions between ROIs.

### Vascular density

ROIs were manually outlined around each anatomical subregion according to the Allen Brain Atlas, and using visible anatomical landmarks. Due to differences in fluorescent labeling intensity, each region was thresholded individually to isolate labeled blood vessels from the background, so each region measured from 0.023 to 4.67 mm² within each hemisphere/section. Area fraction of blood vessels above the threshold was measured for each ROI. In total, mean vascular density was calculated from multiple subregions for 17 ROI (olfactory area - OLF; cingulate cortex - CA; retrosplenial cortex

- RSP; primary visual - V1; primary somatosensory - S1; primary motor area - M1; auditory - AUD; hippocampus - HIP; perirhinal - PERI; insular - INS; thalamus - TH; habenula - HAB; hypothalamus - HY; caudate putamen - CP; white matter - WM; pericisternal - PCS; ependymal around lateral ventricles - EPD) in 6 KO and 6 WT animals. Further statistical comparison was performed assuming inhomogeneous signal distribution properties between different ROI (similarly as for DWI). Hence, considering independent measurements of vascular densities among ROI analyzed and due to small group size, nonparametric Mann-Whitney U-test was employed to compare the vessel densities from KO and WT animals ROI-wise.

For ROI-wise correlation analysis, a mean value of AQP4 expressions as well as vascular densities at ROI was calculated from all respective animals strain-wise.

### TMA measurements *in vivo* and remaining statistics

For ISF space volume estimation with TMA: to keep uniformity of obtained α and $\lambda$ values distribution within KO and WT groups altered by uneven group size, only the animals expressing α within mean ± 1.5 × SD of α distribution within the group (within ~90% of cumulative distribution) were kept. Therefore, two animals from KO and three animals from WT group were removed from among awake batch, and one KO and four WTs were removed from the K/X anesthetized batch of animals (6–7 KO and 16–17 WT remaining).

The obtained α and $\lambda$ values form TMA were compared using one-way ANOVA with Bonferroni's post-hoc. Differences in α and $\lambda$ values between awake and anesthetized animals, total excreted CSF volumes, brain water contents, as well as demographic characteristics parameters (*Table 1A*) between KO and WT mice were compared using Mann-Whiteny U-test.

### Visualizations

Whisker-box and correlation plots were generated using Graphpad and radar plots comparing DWI and IVIM values ROI-wise were generated using OriginPro (v. 2020, OriginLab Corporation, Northampton, MA).

For the purpose of 3D-CISS volumetry and cisternography figures plotting or to visually depict the CSF signal changes within the DCE-MRI time-series, all 3D maximum intensity projection and multiplanar reconstruction images were generated using Amira version 6.2 (Thermo Fisher Scientific, Waltham, MA, USA). 3D-CISS surface reconstructions were performed using Mango Image Analysis software (v.4.1, Research Imaging institute, UTHSCA).

To representatively visualize both AQP4 channel and vascular immunohistochemistry staining, a confocal microscopy (objective UplanXApo 60x / numerical aperture 1.42, ∞ / compatible cover glass thickness 0.17 mm / field number 26.5 mm, oil immersion; Olympus) was performed in a representative section from a single WT animal. Single FOVs were acquired in a 401×401 px / 166.14×166.13 μm frames (0.172 μm$^2$/pixel) and overlaid in ImageJ to visualize both the position of AQP4 channels as well as the vasculature in a representative image.

## Acknowledgements

This study was supported by Lundbeck Foundation (R359-2021-165), Novo Nordisk Foundation (NNF20OC0066419), and the Dr. Miriam and Sheldon G Adelson Medical Research Foundation. National Institutes of Health grants (R01AT011439 and U19NS128613), The U.S. Army Research Office MURI (W911NF1910280), Human Frontier Science Program (RGP0036) and The Simons Foundation (811237). The views and conclusions contained in this article are solely those of the authors and should not be interpreted as representing the official policies, either expressed or implied, of the National Institutes of Health, the Army Research Office, or the U.S. Government. The U.S. Government is authorized to reproduce and distribute reprints for Government purposes notwithstanding any copyright notation herein. The funding agencies have taken no part on the design of the study, data collection, analysis, interpretation, or in writing of the manuscript.

We thank Dan Xue for assistance with the illustrations, and Palle Koch for help with manufacturing MRI phantoms.

# Additional information

## Funding

| Funder | Grant reference number | Author |
|---|---|---|
| Lundbeckfonden | | Maiken Nedergaard |
| Novo Nordisk Fonden | | Maiken Nedergaard |
| National Institutes of Health | | Maiken Nedergaard |
| Army Research Office | | Maiken Nedergaard |
| Human Frontier Science Program | | Maiken Nedergaard |
| Simons Foundation | | Maiken Nedergaard |
| Adelson Family Foundation | | Maiken Nedergaard |

The funders had no role in study design, data collection and interpretation, or the decision to submit the work for publication.

## Author contributions

Ryszard Stefan Gomolka, Conceptualization, Software, Formal analysis, Validation, Investigation, Visualization, Methodology, Writing - original draft, Writing – review and editing; Lauren M Hablitz, Formal analysis, Supervision, Validation, Investigation, Methodology, Writing – review and editing; Humberto Mestre, Formal analysis, Validation, Investigation, Methodology, Writing – review and editing; Michael Giannetto, Formal analysis, Validation, Investigation, Visualization; Ting Du, Formal analysis, Validation, Investigation; Natalie Linea Hauglund, Resources, Validation, Investigation; Lulu Xie, Weiguo Peng, Paula Melero Martinez, Validation, Investigation; Maiken Nedergaard, Conceptualization, Supervision, Funding acquisition, Methodology, Project administration, Writing – review and editing; Yuki Mori, Conceptualization, Supervision, Investigation, Visualization, Methodology, Project administration, Writing – review and editing

## Author ORCIDs

Ryszard Stefan Gomolka http://orcid.org/0000-0001-9797-1062
Lauren M Hablitz http://orcid.org/0000-0001-6159-7742
Humberto Mestre http://orcid.org/0000-0001-5876-5397
Michael Giannetto http://orcid.org/0000-0002-4338-8709
Natalie Linea Hauglund http://orcid.org/0000-0002-2198-6329
Maiken Nedergaard http://orcid.org/0000-0001-6502-6031
Yuki Mori http://orcid.org/0000-0003-4208-0005

## Ethics

All experiments were performed based on approval received from both the Danish Animal Experiments Inspectorate (License number: 2020-15-0201-00581) and the University of Rochester Medical Center Committee on Animal Resources (UCAR, Protocol 2011-023).

## Decision letter and Author response

Decision letter https://doi.org/10.7554/eLife.82232.sa1
Author response https://doi.org/10.7554/eLife.82232.sa2

# Additional files

## Supplementary files
• MDAR checklist
• Source data 1. Aggregated data set, including all source data.

## Data availability

Entire data from the paper is available in the .xls data file attached. The attached data file is subdivided into separate sheets, each for a single experiment and accompanied with respective heading and descriptions, and provides the possibility of replicating all figures and statistics. A summary of data is presented in the tables and figures within the paper. A detailed description of an author algorithm for CSF space segmentation from 3D-CISS images, as well as DWI analysis, is provided in the Materials and Methods section (page 14 onward).

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

# Appendix 1

## Supplementary correlational analysis

To associate the findings from MR-DWI, a correlation between the mean AQP4 expression and mean ADC and D from the measured parenchymal ROI was computed for WTs. Additionally, to confirm whether ADC and D may reflect the free fluid volume, and thus the ISF space volume in the brain, Pearson's linear correlation was calculated between the mean DWI and IVIM, and T1 values and free water volumes fractions from the phantoms.

Similarly, the estimated mean vascular densities were correlated to the mean ADC and IVIM diffusion values and to the MR tracer dynamics scores ROI-wise, for both KO and WT separately. In search for potential relationships, correlation was computed further between AQP4 expression and the vascular density as well as between MR diffusion and MR tracer dynamics parameters ROI-wise. The correlations were calculated only for the set of parenchymal ROI assessed with both compared methods, and considered significant for the highest value of linear Pearson's r- or range Spearman's rho-correlation>50%, with pP<0.05 and non-zero slope of the regression line.

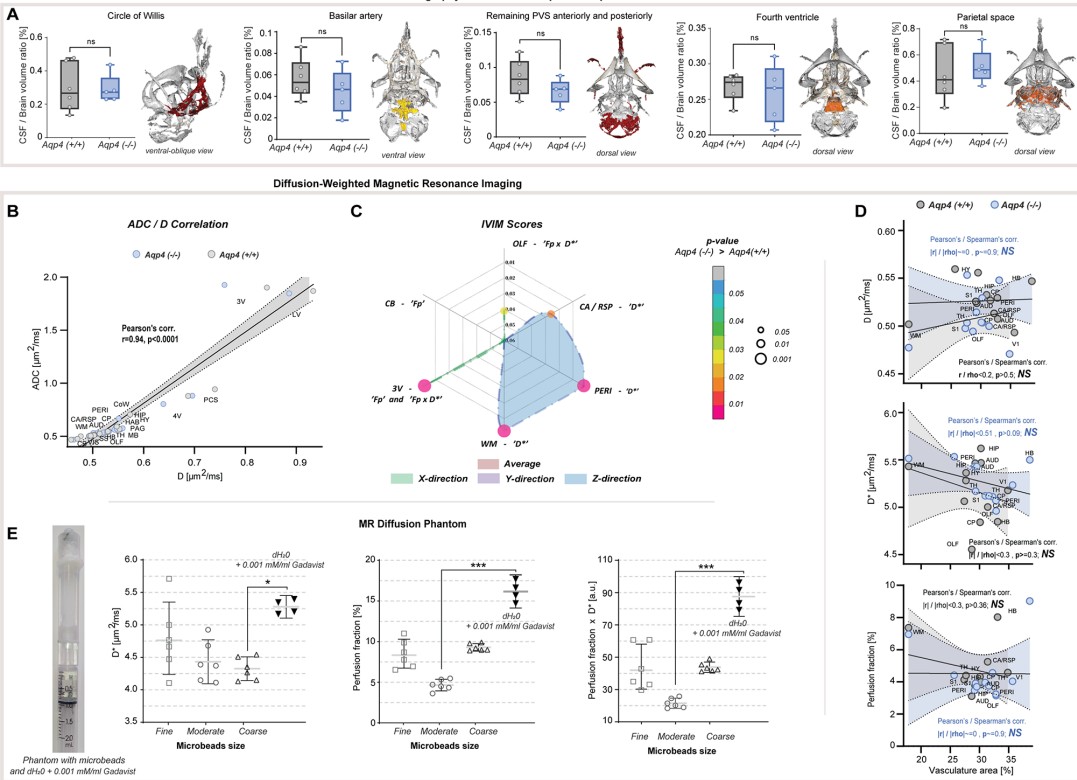

**Appendix 1—figure 1.** Supplementary results for MR CSF space volumetry and MR diffusion evaluation. (**A**) Whiskers-box plot comparison of segmented CSF regions, where no statistical difference between KO and WT was found, along with overlaid 3D surface images of separate compartments (colored) from exemplary WT animal: CSF space at the skull base - Circle of Willis; perivascular space (PVS) around the basilar artery and its branches; PVS anteriorly and posteriorly, remaining after separating the ventricular and the space at the skull base; CSF space in the fourth ventricle; parietal PVS and cisterns (*Figure 1—source data 1*). (**B**) Correlation plot for ADC and D slow diffusion obtained from 6 AQP4 KO and 6 WT animals analyzed. (**C**) Radar plot showing statistical significances for the differences between IVIM scores, found among 5 parenchymal and 1 CSF space ROI assessed, for average and in 3 diffusion-encoding directions jointly (*Figure 2—source data 1*). (**D**) Correlation plots for ROI-wise comparison between IVIM measures and vascular densities from 12 parenchymal ROI analyzed with both methods in KO and WT animals, along with calculated linear and range correlation scores. (**E**) Fast diffusion IVIM measures calculated in a distilled water phantom (+0.001 mM/ml gadobutrol) and water phantoms filled with Sephadex-G25 microbeads of fine, moderate, and coarse sizes (*Figure 2—source data 2*). **Legend:** OLF-olfactory, CA / RSP-cingulate / retrosplenial, VIS (**V1**)-visual, SS (**S1**)-somatosensory, AUD-auditory, HIP-hippocampus, PERI-perirhinal, TH-thalamus, HAB-habenula, HY-hypothalamus, MB-midbrain, PAG-periaqueductal gray, HB-hindbrain; CP-
*Appendix 1—figure 1 continued on next page*

*Appendix 1—figure 1 continued*

caudate putamen, WM-white matter; 3V-third ventricle, LV-lateral ventricle, 4V-fourth ventricle, PCS-pericisternal space, CB-cerebellum, D*-pseudodiffusion (fast diffusion), $F_P$-perfusion fraction; ns-not significant, *-p<0.05, **-p<0.01, by means of Mann-Whitney U-test (**A, C**), Kruskal-Wallis one-way ANOVA with Dunn's correction (**E**). All correlation plots show respective regression lines along with semi-transparent areas marking 95% confidence intervals of the fitting. The highest obtained Pearson's linear or Spearman's range correlation scores reported and considered significant if correlation value >0.5 with *P*<0.05, and non-zero regression slope.

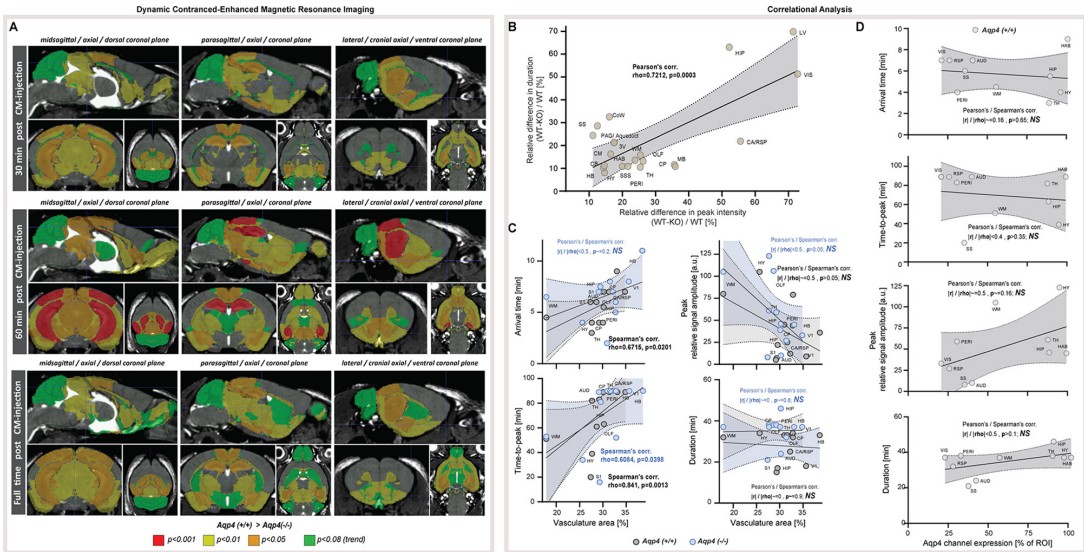

**Appendix 1—figure 2.** Supplementary results for dynamic contrast-enhanced MRI *in vivo* and correlational analysis. (**A**) 3D multiplanar reconstructions of dynamic-contract enhanced (DCE) MRI – midsagittal, parasagittal, and lateral along with orthogonal axial and coronal slices from mean 3D FISP images from 5 AQP4 KO and 6 WT along with segmentation maps color-coded according to p-significance value from nonparametric Two-way ANOVA with post-hoc, for 3 time points: 30 minutes (top), 60 minutes (middle) and 90 minutes (full time, bottom) after applying gadobutrol injection via cisterna magna (CM) (*Figure 3—source data 1*). (**B**) Correlation plot for the relative duration and peak intensity differences between AQP4 KO and WT animals analyzed. (**C**) Correlation plots between DCE-derived scores and the vascular densities from 12 parenchymal ROI analyzed with both methods in AQP4 KO and WT animals. (**D**) Correlation plots between DCE-derived scores and AQP4 expression from 10 parenchymal ROI analyzed with both methods in AQP4 KO and WT animals. **Legend:** OLF-olfactory, CA / RSP-cingulate / retrosplenial, VIS (**V1**)-visual, SS (**S1**)-somatosensory, AUD-auditory, HIP-hippocampus, PERI-perirhinal, TH-thalamus, HAB-habenula, HY-hypothalamus, MB-midbrain, PAG-periaqueductal gray, HB-hindbrain; CP-caudate putamen, WM-white matter; 3V-third ventricle, LV-lateral ventricle, CoW-Circle of Willis, CM-cisterna magna, SSS- superior sagittal sinus; *NS*-not significant. All correlation plots show respective regression lines along with semi-transparent areas marking 95% confidence intervals of the fitting. The highest obtained Pearson's linear or Spearman's range correlation scores reported and considered significant if correlation value >0.5 with *P*<0.05, and non-zero regression slope.

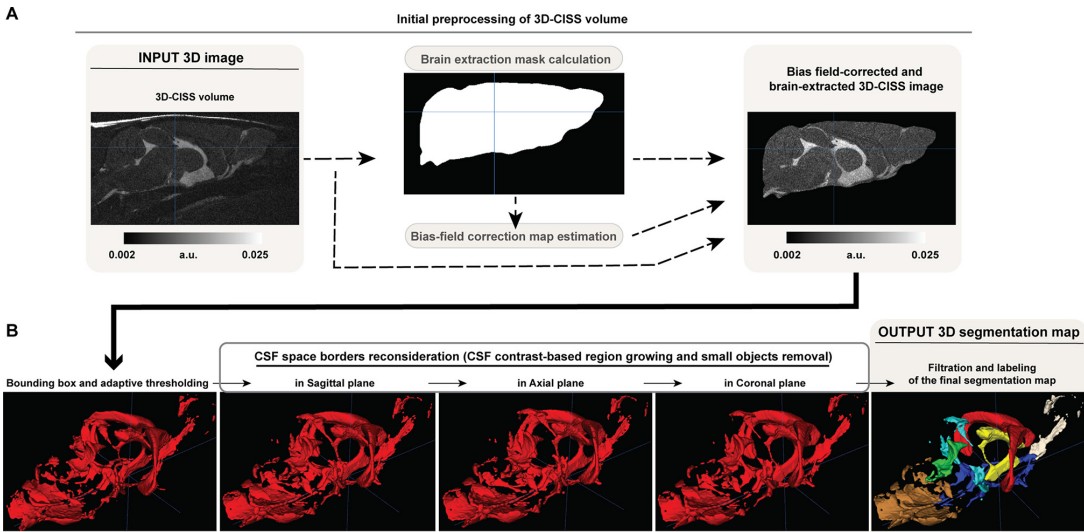

**Appendix 1—figure 3.** Schematic flow chart of the algorithm for the 3D-CISS-based CSF space volumetry and cisternography in *Aqp4(-/-) (KO)* and *Aqp4(+/+) (WT)* mice, including initial preprocessing (bias field correction and brain extraction) of the computed 3D-CISS volume (**A**) and subsequent automatic CSF space segmentation (**B**), based on representative 3D-CISS volume from a single animal.

