## [Editor Report]

This important investigation is of interest to neuroimaging scientists and neurophysiologists studying the glymphatic system. Using a multi-modal approach including magnetic resonance and histological methods, this work provides substantial data interrogating the effect of removing aquaporin-4 (AQP4) from the mouse brain parenchyma on the structural morphology and interstitial fluid dynamics stagnation. In particular, the authors provide convincing evidence that deletion of AQP4 in mice results in increased interstitial volume, likely due to increased resistance to parenchymal CSF efflux.

---

## [Decision Letter]

**Decision letter after peer review:**

Thank you for submitting your article "Loss of aquaporin-4 results in glymphatic system dysfunction via brain-wide interstitial fluid stagnation" for consideration by *eLife*. Your article has been reviewed by 3 peer reviewers, and the evaluation has been overseen by a Reviewing Editor and Ma-Li Wong as the Senior Editor. The following individual involved in the review of your submission has agreed to reveal their identity: Yolanda Ohene (Reviewer #3).

Essential revisions:

Reviewer 1 notes the difficulty in distinguishing between the direct effects of the lack of aquaporin-4 channels (acute) vs the developmental effects of the AQP4 knockout (chronic). While it may not be necessary to conduct new experiments, it is worth discussing this shortcoming of interpretation in detail.

Reviewer 2 suggests some potential further analyses that may strengthen the paper. If these are not feasible, the issues they aim to clarify should be discussed.

Reviewer 3 is unconvinced by the speculation regarding the upregulation of AQP4.

The reviewers request some additional information including:

– details of histological imaging.

– regional brain volume estimates.

– pseudo-diffusion estimates.

In addition, the reviewers request discussion on a number of key points as detailed by individual reviewers:

– para- vs trans-cellular flow;

– amount vs location of AQP4 channel expression in knockouts;

– potential effects of edema;

– interpretation of TGN020 agent;

– sensitivity of diffusion MRI to exchange vs restriction;

– limitations of IVIM modelling.

*Reviewer #1 (Recommendations for the authors):*

There is a number of points that would need to be addressed in order to improve the quality of this paper before it can be accepted for publication:

– It's important to determine whether the differences in the AQP4 -/- are due directly to the lack of AQP4 per se, or due to the fact that the CNS has developed in the absence of AQP4 – this could be determined by overexpressing AQP4 using a glial-specific virus in the AQP4 -/- animals and/or knocking down AQP4 in WT animals using si/shRNA in a similar virus. This can delineate differences between acute and chronic loss of AQP4.

– Imaging was an essential aspect of the ex vivo part. Authors need to provide more details such as how many FOVs have been taken and what are their measures to minimize biases, and how they have excluded any possible interference from background signals in order to enhance the reproducibility of the presented data. Magnification numbers should be included for all the figures. But it won't be enough as it has nothing to do with resolution, especially for the purpose of quantitative analyses like in this study. So, the authors need to include the NA of the utilized lens as well.

– Line 203-204 "Next, to identify possible differences in the tracer dynamics between KO and WT mice, we calculated arrival time, time‐to‐peak, peak intensity, and duration of significant from baseline parenchymal tracer accumulation (Table 3)". The field tends to discuss 'water' and 'fluid' in a manner that incorrectly suggests their interchangeability. However, it is important to note that in the glymphatic system, the clearance of brain waste occurs through paracellular flow. Classic tracer studies measure this paracellular flow, while the use of H217O captures both paracellular flow and diffusive transcellular exchange of water. Importantly, both are AQP4 dependent – one directly and one indirectly. This needs to be clarified. References:

https://www.nature.com/articles/s41583-021-00514-z

https://academic.oup.com/brain/article/145/1/64/6367770

– Figure 2I and related discussion: the increased AQP expression and the redistribution/surface localization can be two different concepts. Previous studies have shown an increase in AQP4 membrane localisation in human astrocytes which wasn't accompanied by a change in AQP4 protein expression levels. References:

https://www.ncbi.nlm.nih.gov/pmc/articles/PMC5765450/

https://www.ncbi.nlm.nih.gov/pubmed/31242419

https://pubmed.ncbi.nlm.nih.gov/26013827/

To this extent, the authors omit a key study from Kitchen et al. 2020 demonstrating that the development of edema following injury-induced hypoxia is AQP4 dependent. That study shows that CNS edema is associated with increases both in total aquaporin-4 expression and aquaporin-4 subcellular translocation to the blood-spinal cord barrier (BSCB). Pharmacological inhibition of AQP-4 translocation to the BSCB prevents the development of CNS edema and promotes functional recovery in injured rats. This role has been recently confirmed by the work of Sylvain et al. BBA 2021 which has demonstrated that targeting AQP4 effectively reduces cerebral edema during the early acute phase of stroke using photothrombotic stroke model. References:

https://www.cell.com/cell/fulltext/S0092-8674(20)30330-5.

https://pubmed.ncbi.nlm.nih.gov/33561476/

*Reviewer #2 (Recommendations for the authors):*

– One concern is the use of the term "exchange". It is reasonable to extrapolate that decreased CSF efflux and transmembrane water exchange are what lead to increased interstitial space volume, measured here via 3D-CISS + segmentation. However, this does not imply that the diffusion metrics are altered directly due to decreased transmembrane exchange. Rather, depending on the diffusion times probed, the reduced restriction of the enlarged interstitial space may be the dominant effect on the signal, with little to no transmembrane water exchange during the encoding. The authors should be careful to distinguish between exchange as a causal mechanism for changes in steady-state brain water and interstitial volume as opposed to exchange as a mechanism giving rise to the diffusion signal.

– Related to the above, there may be an opportunity to provide a quantitative look at transmembrane water exchange. Provided that CSF production was similar between WT and KO, the total change in interstitial volume could be related to a decrease in the apparent exchange rate by assuming a simple first-order rate model of exchange. That is, where the total CSF efflux is proportional to the surface-to-volume ratio of the CSF space and the exchange rate or permeability. One could predict the decrease in the exchange rate (as stated in the discussion: "increased resistance towards… efflux") that leads to a given change in total volume, assuming constant surface area. Direct comparisons could then be made, e.g., to the findings concerning exchange rates in Ref. 22. Including such an analysis, though speculative, may strengthen the conclusions of the paper and provide additional points of comparison to the literature.

– The use of an IVIM signal model is well-motivated by the comparison to vascular density as quantified by histological staining. However, the biological validity of IVIM and bi-exponential signal models in tissue, particularly GM (see recent work on SANDI and other GM signal models for DWI), remains contentious. To help validate the use of IVIM metrics beyond the comparison to standard ADC fitting provided here, an additional analysis of the distribution of apparent diffusivities using an inverse Laplace transform is suggested. This analysis could be carried out only in ROIs where changes in IVIM metrics were observed (namely the olfactory area) to validate (i) a bimodal distribution of diffusivities and (ii) the approximate fractions of slow/fast diffusivities.

– Related to the above, IVIM metrics can vary depending on the TE and/or gradient amplitude, which may make comparisons between IVIM studies difficult. See, for instance: 10.1002/mrm.28996. This should be mentioned or addressed.

– On line 289, the authors discuss that WT expression of AQP4 is localized to areas of enlarged interstitial volume and suggest that this may be the result of a compensatory mechanism. References to recent studies of dynamic relocalization of AQP4 and some elaboration concerning the potential signalling pathway would strengthen this suggestion – e.g., see Salman et al. (2022) https://doi.org/10.1093/brain/awab311

– For the DCE results, the authors argue that the shorter duration of interstitial tracer accumulation in KO vs. WT can be explained by decreased tracer penetration into the brain. Perhaps this could be quantified by looking at the cumulative area under the curve vs. time as a proximal measurement of tracer influx/efflux (although the use of arbitrary intensity units may complicate such an analysis). The data is convincing as presented, but there may be a more succinct visualization/analysis that reveals impaired tracer influx and efflux in the KO mice.

– In some of the ROIs, the DCE time series appears to show two peaks (Figure 3: cisterna magna, Circle of Willis, superior sagittal sinus). Is there any significance or interpretation of the shape of these time series?

– I understand that the CSF segmentation algorithm was developed in-house and will be evaluated and presented in full in a later technical report. At this stage, it may nonetheless be helpful to provide the output of each step in the segmentation process as a supplementary figure – at least for a single ROI.

*Reviewer #3 (Recommendations for the authors):*

1. Figure 2 is very dense, making it difficult for the reader to interpret results, authors should consider separating the diffusion-weighted MRI section from the histology section to improve readability.

2. The ADC values presented in Figure 2C are lower than ADC generally measured in the mouse brain, could the author comment on why they see an underestimation in their work?

3. Authors could remove the MRI diffusion phantom work from Figure 2 and include it as an additional Supplementary figure. While it complements the work, it is fairly well established that ADC increase is related to the extravascular space.

4. Could authors include regional brain volume measurements?

5. Page 11, Line 238: "possible reflecting that AQP4 is upregulated in response to stagnation of interstitial fluid" may be too strong a speculation as an increase in ADC is likely suggesting a higher rate of fluid movement. And again on page 13, line 289: "compensatory upregulation of AQP4 to fluid stagnation consistent with the notion that AQP4 reduce parenchymal resistance to water and solute movement" seems contradictory as AQP4 water channel is known as the path of least resistance for water transfer (Manley et al., Nature Medicine, 2000; Papadopoulos et al., Faseb J, 2004), therefore it does not follow that an upregulation would lead to fluid stagnation.

6. Page 14, Line 312: "Instead, we show that slow diffusion of water across physiological membrane is increased due to more water exchange in an enlarged interstitial space" could be more precise. For example, previous work has shown that blood-brain barrier water exchange is reduced due to the removal of AQP4 (Urshihata et al. Eur Radiol Exp, 2021; Ohene et al., Neuroimage, 2019).

7. Would the authors be able to provide values for the pseudo-diffusion (D*) measurements?

---

## [Author Response]

Essential revisions:Reviewer 1 notes the difficulty in distinguishing between the direct effects of the lack of aquaporin-4 channels (acute) vs the developmental effects of the AQP4 knockout (chronic). While it may not be necessary to conduct new experiments, it is worth discussing this shortcoming of interpretation in detail.

We agree that genetic knockout compared to acute inhibition of AQP4 may provide different phenotypes, and have further discussed this in the manuscript:

Introduction, page 1, lines 38 to 47: […] AQP4 KO mice exhibit a 25-60% decrease in glymphatic CSF tracer influx (20-22), and acute pharmacological blockade of AQP4 inhibits glymphatic transport using TGN-020 inhibitor (23-25), reducing severity of brain edema and lesion volume after ischemic injury (26, 27). However, in cell-based assays TGN-020 inhibitor failed to inhibit AQP4, bringing into question the true molecular mechanisms of its action (28), discussed in (29). Deletion of AQP4 also accelerates buildup of neurotoxic protein waste in neurodegenerative models of Alzheimer’s (30, 31) and Parkinson’s disease (32). In humans, a common single nucleotide AQP4 polymorphism is correlated to changes in slow-wave non-REM sleep and cognition (33), consistent with increased glymphatic function during sleep (34, 35). Recent reports highlight potential roles of AQP4, especially in edema formation after hypoxia due to the spinal cord injury, and in early and acute phases of stroke (36-38). Thus, the evidence suggests that it is the vascular polarized AQP4 expression in the astrocytic vascular endfeet functionally crucial for fluid transport, and not necessarily total AQP4 levels in the tissue 20, 21,

Reviewer 2 suggests some potential further analyses that may strengthen the paper. If these are not feasible, the issues they aim to clarify should be discussed.

In response to the Reviewer’s remarks, we have provided supplementary analyses.

1) We clarify that we have intentionally analyzed DWI with monoexponential and biexponential fitting approaches to assess mainly the interstitial fluid space in two genotypes. Both approaches possess clinical and translational applicability, and are feasible considering non-isotropic voxel dimension in acquired EPI images (0.15 × 0.15 × 0.5 mm^3^). In response to the remark, we have provided supplementary analysis using inverse Laplace (ILT) transform method to verify the distribution of diffusivities in depicted DW signals. Nevertheless, this analysis requires further supplementary validation including brain-wide assessment and phantom/synthetic signal analysis before its feasibility, in our suboptimal for ILT purpose setup including non-segmented EPI in live animals, can be confirmed. Therefore, we kindly request on keeping the analysis only with the response to the remarks.

**Author response image 1. sa2fig1:** Inverse Laplace transform (ILT) performed using INSPECT algorithm assuming presence of 2 spectral components. Presences of fast diffusivities (>10 um^2^/ms, perfusion-like) components was confirmed in aggregated ADC spectra from 3 regions analyzed (A-C, see ‘Distribution diffusion-encoding direction-wise’). Although contribution of fast diffusivities is minimal to the mean spectrum from all diffusion-encoding directions jointly (‘Mean of aggregated distribution’), its presence confirms applicability of IVIM method for separating the perfusion-related effects from the spectrum of slow diffusivities. Two distinctive compartments in the range of restricted and low range of free water diffusivities were found for all cortical (A) and parenchymal (B) ROI (see ‘Mean of aggregated distribution’ and ‘Averaged distribution’) but not the 3rd ventricular CSF space ROI where an additional peak was visible in the fast diffusivity range (C). As further requested, we have provided comparison for ratios of depicted two diffusivity peaks fractions for analyzed ROI (D). Legend: F_restricted_ – fraction of restricted compartment, F_free_ – fraction of free water compartment, ns-not significant (Mann-Whitney U-test).

Our DWI approach and further possibilities for modelling were additionally discussed in the manuscript:

Materials and methods, page 15, lines 423 to 433: ‘[…] It is worth mentioning, that there is an ongoing debate on efficacy of IVIM modelling in reflecting phenomena in microvascular network or related to tissue microarchitecture (47, 84-87). […] However indicatively useful, higher order models and the models focused on separating hindered MR diffusion signal according to assumption on microarchitecture (94-100) or assessment of diffusion signal distribution (101-104) are beyond presented general evaluation.’

2) Furthermore, we have included a supplementary analysis of DCE-MRI time-curves to better understand contrast injection-related phenomena. The results of the analysis were included into the manuscript.

Materials and methods, page 22, lines 717 to 721: […] Duration of significant from baseline interstitial tracer accumulation was obtained as a time difference between the last and the first time points, where DCE signal increase was significantly different from baseline. Area under the DCE curve was calculated for each subject as a sum of DCE signal amplitudes over all the measured time points. […]

Results, page 8, lines 224 to 225: ‘This would be confirmed with the area under the DCE curve that consistently was smaller in KO (Table 3, ‘Mean AUC’).’

Results, Page 9, lines 247 to 250: […] Finally, there was no correlation found between the vascular density and AQP4 expression (both Person’s and Spearman’s correlation absolute value <0.52 and p>0.16), or between MR diffusion and tracer dynamics parameters (correlation value <0.5 and p>0.05 for any comparison) except for low correlation between the area under the DCE curve and ADC for both genotypes (WT: r=0.63, p=0.0272; KO: r=0.64, p=0.0239 ). […]

Reviewer 3 is unconvinced by the speculation regarding the upregulation of AQP4.

We have edited the manuscript to clarify our rationale for the upregulation of AQP4.

Discussion, page 12, lines 303 to 310:

[…] These correlations suggest that the vascular network provides a highway for perivascular CSF inflow and thereby drives the initial tracer distribution within the parenchyma. Increased AQP4 expression in regions manifesting high ADC or D in WTs, possible due enlarged interstitial volume, may reflect a compensatory upregulation of AQP4 due to fluid stagnation and accordant with the notion that AQPs reduce parenchymal resistance and facilitate the water and solute movement. Consistent with this hypothesis, recent studies report dynamic AQP4 relocalization leading to changes in signaling pathways (36). Therefore, our data overall indicates that the markedly altered brain fluid transport in AQP4 KO mice may result from a reduction in glymphatic fluid export, leading to stagnation of ISF and enlargement of the interstitial space. […]

The reviewers request some additional information including:– Details of histological imaging.

The details of histological imaging are provided in the Materials and methods, as well as in the Figure 2 and its caption.

Materials and methods, page 22, lines 748 to 754: ‘Brain sections were imaged using a conventional fluorescence macroscope (Stereo Investigator with objective UplanXApo 10x/numerical aperture 0.40, ∞/compatible cover glass thickness 0.17 mm/field number 26.5 mm, no immersion liquid (air); Olympus) and subsequently analyzed using ImageJ. Multiple FOVs were acquired in a 1360 × 1024 px / 1392.44 × 1048.43 µm frames (1.048 µm2/pixel), and subsequently aligned and stitched together to provide image of entire brain section. Both AQP4 and vascular staining were assessed in a complimentary anterior and posterior entire brain sections (cf. AQP4 channel staining ex vivo). To minimize the reader-associated bias, all image analyses were performed by the investigator (MG) blinded to the image content and animal genotype, and in randomized images.’

Materials and methods, page 23, lines 802-806: ‘To representatively visualize both AQP4 channel and vascular immunohistochemistry staining, a confocal microscopy (objective UplanXApo 60x / numerical aperture 1.42, ∞ / compatible cover glass thickness 0.17 mm / field number 26.5 mm, oil immersion; Olympus) was performed in a representative section from a single WT animal. Single FOVs were acquired in a 401 × 401 px / 166.14 × 166.13 µm frames (0.172 µm^2^/pixel) and overlaid in ImageJ to visualize both the position of AQP4 channels as well as the vasculature in a representative image.’

– Regional brain volume estimates.

CISS and DWI using EPI are not designed to provide accurate measurements of regional brain morphometry due to low parenchymal contrast or associated magnetization and wrapping / ghosting artifacts. In response to Reviewer’s request, we have provided a supplementary analysis based on affine registration of Allen Brain atlas template (used for DCE-MRI evaluation) subject-wise to available 3D-CISS images. We have found no statistical difference between the regional volumes from *Aqp4 (+/+)* and *Aqp4 (-/-)* mice (Figure below), for all 15 parenchymal regions analyzed in our original manuscript. However, it cannot be excluded that such approach is affected by additional registration-related artefacts and corregistration with common template would assume similar proportionality between the parenchymal subregions in both genotypes. This is a huge caveat, as AQP4 expression is not homogenous within the brain in normal conditions (see Figure 2H). Therefore, we kindly request on keeping the provided morphometry analysis outside the manuscript and only within the responses to the reviewer’s remark.

**Author response image 2. sa2fig2:** 3D-CISS-based regional brain morphometry in *Aqp4* (+/+) and *Aqp4* (-/-) mice brains. Legend: OLF-olfactory, CA-cingulate, VIS (V1)-visual, SS (S1)-somatosensory, M-motor; AUD-auditory, HIP-hippocampus, PERI-perirhinal, TH-thalamus, HAB-habenula, HY-hypothalamus, MB-midbrain, PAG-periaqueductal gray, HB-hindbrain; CP-caudate putamen, WM-white matter; 3V-third ventricle, LV-lateral ventricle, CoW-Circle of Willis, CM-cisterna magna, SSS- superior sagittal sinus; NS-not significant. Minimal p-value (p=0.1385) found for the cingulate area.

To provide further clarification of our MRI protocol purpose, we have renamed the 3D-CISS protocol as ‘CSF space volumetry and cisternography’ in every mention in the manuscript, compared to the previous ‘volumetry and cisternography’ (page 21, line62; page 12, line 293; page 15, line 404, 412 and 413; page 19, line 569; page 24, line 807).

– Pseudo-diffusion estimates.

Pseudo-diffusion estimates along with all DWI-derived parameters were provided in the supplementary.xls file combining all used data. We have additionally referred to this supplementary file in the methods section page 21, line 699: ‘[…] (see calculated values in the supplementary Excel file ‘Figure 1 – Source data 1’) […]’.

In addition, the reviewers request discussion on a number of key points as detailed by individual reviewers:– para- vs trans-cellular flow;– amount vs location of AQP4 channel expression in knockouts;– potential effects of edema;– interpretation of TGN020 agent;– sensitivity of diffusion MRI to exchange vs restriction;– limitations of IVIM modelling.

We have included discussion and references for para- vs. trans-cellular flow (Results and Discussion, page 4 and page 13), amount vs. localization of AQP4 (Discussion, page 13), potential effects of edema (Introduction, page 1), interpretation of TGN020 (Introduction, page 1), sensitivity of diffusion MRI to exchange vs. restriction (Discussion and Methods, page 13 and page 15), and limitations of IVIM modeling (Methods, page 15) into the manuscript.

Reviewer #1 (Recommendations for the authors):There is a number of points that would need to be addressed in order to improve the quality of this paper before it can be accepted for publication:– It's important to determine whether the differences in the AQP4 -/- are due directly to the lack of AQP4 per se, or due to the fact that the CNS has developed in the absence of AQP4 – this could be determined by overexpressing AQP4 using a glial-specific virus in the AQP4 -/- animals and/or knocking down AQP4 in WT animals using si/shRNA in a similar virus. This can delineate differences between acute and chronic loss of AQP4.

We agree that global genetic deletion of AQP4 may lead to developmental adaptations that are less than ideal. Alterations in water homeostasis are consistent across AQP4 KO animals, and in α-Syntrophin knockout animals that express mislocalized AQP4 within the brain (Mestre et al. 2018; https://pubmed.ncbi.nlm.nih.gov/30561329/). This suggests that our findings are most likely an AQP4 localization phenotype, as opposed to a genetic fluke.

No one has, to our best knowledge, been able to manipulate AQP4 expression by more than 50-60%, or similar to *Aqp4 (-/+)* heterozygotes. As heterozygotes do not exhibit phenotype (Ma et al. 1997; https://pubmed.ncbi.nlm.nih.gov/9276712/), such small manipulations would not be able to recapitulate the phenotypes seen here. It is worth noting that, as of right now, none of these studies have been published bringing into question the validity of viral or si/shRNA upregulation or knockdown of AQP4 in general. This technical challenge is beyond the scope of our manuscript.

Thus, we have included a discussion of these potential caveats to our model in the manuscript:

Introduction, page 1, lines 38 to 45: […] AQP4 KO mice exhibit a 25-60% decrease in glymphatic CSF tracer influx (20-22), and acute pharmacological blockade of AQP4 inhibits glymphatic transport using TGN-020 inhibitor (23-25), reducing severity of brain edema and lesion volume after ischemic injury (26, 27). However, in cell-based assays TGN-020 inhibitor failed to inhibit AQP4, bringing into question the true molecular mechanisms of its action (28), discussed in (29). Deletion of AQP4 also accelerates buildup of neurotoxic protein waste in neurodegenerative models of Alzheimer’s (30, 31) and Parkinson’s disease (32). In humans, a common single nucleotide AQP4 polymorphism is correlated to changes in slow-wave non-REM sleep and cognition (33), consistent with increased glymphatic function during sleep (34, 35). Recent reports highlight potential roles of AQP4, especially in edema formation after hypoxia due to the spinal cord injury, and in early and acute phases of stroke (36-38). […]’

– Imaging was an essential aspect of the ex vivo part. Authors need to provide more details such as how many FOVs have been taken and what are their measures to minimize biases, and how they have excluded any possible interference from background signals in order to enhance the reproducibility of the presented data. Magnification numbers should be included for all the figures. But it won't be enough as it has nothing to do with resolution, especially for the purpose of quantitative analyses like in this study. So, the authors need to include the NA of the utilized lens as well.

We apologize for the omission. We have edited our Materials and methods section and caption to the Figure 2, and have edited panels F and H of the Figure 2 to improve clarity.

Materials and methods, page 22, lines 748 to 754: ‘Brain sections were imaged using a conventional fluorescence macroscope (Stereo Investigator with objective UplanXApo 10x/numerical aperture 0.40, ∞/compatible cover glass thickness 0.17 mm/field number 26.5 mm, no immersion liquid (air); Olympus) and subsequently analyzed using ImageJ. Multiple FOVs were acquired in a 1360 × 1024 px / 1392.44 × 1048.43 µm frames (1.048 µm2/pixel), and subsequently aligned and stitched together to provide image of entire brain section. Both AQP4 and vascular staining were assessed in a complimentary anterior and posterior entire brain sections (cf. AQP4 channel staining ex vivo). To minimize the reader-associated bias, all image analyses were performed by the investigator (MG) blinded to the image content and animal genotype, and in randomized images.

AQP4 staining. For each brain hemisphere in each section, all ROIs were manually drawn according to the Allen Brain Atlas and using auto fluorescence from the green channel following visible anatomical landmarks and confirming AQP4 expression visibility. Each subregion measured from 0.03 to 1.69 mm^2^. Therefore, a universal threshold was applied manually to all images as a preferred method for reduction of the background signal influence. Subsequently, a mean area fraction covered by AQP4 in both hemispheres/sections was measured for each ROI. In total, mean AQP4 expression was calculated for 11 ROIs (retrosplenial cortex – RSP, visual – VIS; somatosensory – SS; auditory – AUD; hippocampus – HIP; perirhinal – PERI; thalamus – TH; habenula – HAB; hypothalamus – HY; pericisternal – PCS; white matter – WM) in 4 WT mice. […]’

Figure 2 caption: ‘(F) Exemplary immunohistochemistry images (Olympus UplanXApo 10x/numerical aperture 0.40, ∞/compatible cover glass thickness 0.17 mm/field number 26.5 mm, no immersion liquid) (left) […] (Olympus UplanXApo 60x/numerical aperture 1.42, ∞/compatible cover glass thickness 0.17 mm/field number 26.5 mm, oil immersion) (left) […].’

– Line 203-204 "Next, to identify possible differences in the tracer dynamics between KO and WT mice, we calculated arrival time, time‐to‐peak, peak intensity, and duration of significant from baseline parenchymal tracer accumulation (Table 3)". The field tends to discuss 'water' and 'fluid' in a manner that incorrectly suggests their interchangeability. However, it is important to note that in the glymphatic system, the clearance of brain waste occurs through paracellular flow. Classic tracer studies measure this paracellular flow, while the use of H217O captures both paracellular flow and diffusive transcellular exchange of water. Importantly, both are AQP4 dependent – one directly and one indirectly. This needs to be clarified. References:https://www.nature.com/articles/s41583-021-00514-zhttps://academic.oup.com/brain/article/145/1/64/6367770

We have included the following text in the manuscript:

Results, page 8, lines 195 to 199: As of particular importance for studying AQP4, it is worth noting that tracer transport (here gadobutrol) does not directly reflect the movement of water. The water can move into the tissue not only through the paracellular gap between astrocytic endfeet but also via diffusive transcellular exchange. The transport of membrane-impermeable CSF tracers, however, is limited to paracellular transport between the gaps of astrocytic endfeet (Salman et al. 2021, https://pubmed.ncbi.nlm.nih.gov/34408336/; Salman et al. 2022,https://pubmed.ncbi.nlm.nih.gov/34499128/).’

– Figure 2I and related discussion: the increased AQP expression and the redistribution/surface localization can be two different concepts. Previous studies have shown an increase in AQP4 membrane localisation in human astrocytes which wasn't accompanied by a change in AQP4 protein expression levels. References:https://www.ncbi.nlm.nih.gov/pmc/articles/PMC5765450/https://www.ncbi.nlm.nih.gov/pubmed/31242419https://pubmed.ncbi.nlm.nih.gov/26013827/To this extent, the authors omit a key study from Kitchen et al. 2020 demonstrating that the development of edema following injury-induced hypoxia is AQP4 dependent. That study shows that CNS edema is associated with increases both in total aquaporin-4 expression and aquaporin-4 subcellular translocation to the blood-spinal cord barrier (BSCB). Pharmacological inhibition of AQP-4 translocation to the BSCB prevents the development of CNS edema and promotes functional recovery in injured rats. This role has been recently confirmed by the work of Sylvain et al. BBA 2021 which has demonstrated that targeting AQP4 effectively reduces cerebral edema during the early acute phase of stroke using photothrombotic stroke model. References:https://www.cell.com/cell/fulltext/S0092-8674(20)30330-5.https://pubmed.ncbi.nlm.nih.gov/33561476/

We agree that increased AQP4 expression and increased AQP4 polarization are two different concepts, and we have already published on this in vivo (Hablitz Nat Commun 2020, https://pubmed.ncbi.nlm.nih.gov/32879313/), where day/night differences in glymphatic function were found associated with increased localization but not expression of AQP4. Therefore, we have included this point in our manuscript:

Introduction, page 1, lines 44 to 47:

‘Recent reports highlight potential roles of AQP4, especially in edema formation after hypoxia due to the spinal cord injury, and in early and acute phases of stroke (Salman et al. 2017, https://www.ncbi.nlm.nih.gov/pmc/articles/PMC5765450/; Kitchen et al. 2020, https://pubmed.ncbi.nlm.nih.gov/32413299/; Sylvain et al. 2021, https://pubmed.ncbi.nlm.nih.gov/33561476/). Thus, the evidence suggests that it is the vascular polarized AQP4 expression in the astrocytic vascular endfeet functionally crucial for fluid transport, and not necessarily total AQP4 levels in the tissue (Sylvain et al. 2021, https://pubmed.ncbi.nlm.nih.gov/33561476/; Amiry-Moghaddam et al. 2003, https://pubmed.ncbi.nlm.nih.gov/12578959/; Mestre et al. 2018, https://www.ncbi.nlm.nih.gov/pubmed/30561329; Hablitz et al. 2020 https://pubmed.ncbi.nlm.nih.gov/32879313/; Eide and Hansson 2018, https://pubmed.ncbi.nlm.nih.gov/28627088/).’

Reviewer #2 (Recommendations for the authors):– One concern is the use of the term "exchange". It is reasonable to extrapolate that decreased CSF efflux and transmembrane water exchange are what lead to increased interstitial space volume, measured here via 3D-CISS + segmentation. However, this does not imply that the diffusion metrics are altered directly due to decreased transmembrane exchange. Rather, depending on the diffusion times probed, the reduced restriction of the enlarged interstitial space may be the dominant effect on the signal, with little to no transmembrane water exchange during the encoding. The authors should be careful to distinguish between exchange as a causal mechanism for changes in steady-state brain water and interstitial volume as opposed to exchange as a mechanism giving rise to the diffusion signal.

We are thankful for the detailed review and important remarks. Indeed, we propose that the higher slow MR diffusion is reflecting the signal from water protons within increased interstitial space, rather the transmembrane water exchange in AQP4 KO mice. This is supported with our vasculature assessment and good agreement between ADC and D-IVIM findings. Still, we cannot exclude that the increased ISF space might result from increased transmembrane resistance. To improve clarity, we have made the following changes:

– the word ‘transmembrane’ was removed from the results subtitle page 4, line 95

‘Genetic loss of AQP4 alters water diffusivity independent of the microvascular density’

– Page 4, lines 110-112 was rephrased from ‘fast bulk exchange’ to ‘fast bulk water displacement’

‘To evaluate whether a fast bulk displacement of intravascular water protons due to capillary perfusion may contribute to our findings, we performed an additional scoring for differences associated with intra-voxel pseudodiffusive fluid regimes (Le Bihan and Iima 2015, https://pubmed.ncbi.nlm.nih.gov/26204162/), using IVIM diffusion model.’

– Page 4, lines 142-144 was rephrased from ‘transmembrane fluid’ to ‘water passage’

‘This might reflect possible differences in the rate of water passage orthogonally to the differences in associated fluid perfusion markers. Differences in Fp and Fp × D* were found only in the olfactory area and within the 3rd ventricle (min. p<0.05; Table 2B), […]’

– Page 4, lines 146-148 were rephrased to highlight superposition of two contradictory effects (increase in transmembrane resistance and increase in ISF space volume) leading to overall only slight increase in ADC (cf. Pavlin et al. 2017, https://pubmed.ncbi.nlm.nih.gov/28039592/; Urushihata et al. 2021, https://pubmed.ncbi.nlm.nih.gov/34617156/)

‘Overall, slightly increased average ADC and D along with no difference in average IVIM measures suggest existence of larger interstitial space volume in AQP4 KO, perhaps as a result of increased water exchange time (Urushihata et al. 2021, https://pubmed.ncbi.nlm.nih.gov/34617156/), and without tangible alterations in parenchymal blood perfusion.’

– Page 13, lines 333-340 were corrected and expanded with relevant descriptions

‘Instead, we show that slow MR diffusion measures are increased mostly due to an enlarged interstitial space. Only slight 5-6% increase in the mean ADC and D might result from superposition of opposing effects of reduced transmembrane permeability (reducing ADC) and increased extracellular space (increasing ADC) as concluded previously using time-dependent diffusion MRI and Latour’s model of long-time diffusion behavior (Pavlin et al. 2017, https://pubmed.ncbi.nlm.nih.gov/28039592/). Similarly, evaluation of ADC using multi-b-value-multi-diffusion-time DWI provided higher ADC’s in healthy hemispheres of mice subjected to contralateral ischemic stroke, reflecting larger interstitial space in KO (Urushihata et al. 2021, https://pubmed.ncbi.nlm.nih.gov/34617156/). The enlarged interstitial space in both awake and anesthetized AQP4 KO mice is also consistent with previous reports under anesthesia (Yao et al. 2008, https://pubmed.ncbi.nlm.nih.gov/18495879/; Amiry-Moghaddam et al. 2003, https://pubmed.ncbi.nlm.nih.gov/14597704/; Papadopoulos et al. 2004, https://pubmed.ncbi.nlm.nih.gov/15208268/). Our findings in the water phantom filled with Sephadex microbeads of similar porosity but different sizes also confirmed increase in both ADC and D resulting from increased free fluid volume (Figure 2D).’

– Related to the above, there may be an opportunity to provide a quantitative look at transmembrane water exchange. Provided that CSF production was similar between WT and KO, the total change in interstitial volume could be related to a decrease in the apparent exchange rate by assuming a simple first-order rate model of exchange. That is, where the total CSF efflux is proportional to the surface-to-volume ratio of the CSF space and the exchange rate or permeability. One could predict the decrease in the exchange rate (as stated in the discussion: "increased resistance towards… efflux") that leads to a given change in total volume, assuming constant surface area. Direct comparisons could then be made, e.g., to the findings concerning exchange rates in Ref. 22. Including such an analysis, though speculative, may strengthen the conclusions of the paper and provide additional points of comparison to the literature.

Though such a model might be informative, we would like to highlight that not all CSF enters the glymphatic system since there are several direct egress pathways from the skull such as subarachnoid CSF into parasagittal dura (PSD), cribriform, etc. (Rasmussen MK et al. 2018, https://pubmed.ncbi.nlm.nih.gov/30353860/; Ringstad G, et al. 2020, https://pubmed.ncbi.nlm.nih.gov/31953399/). These efflux routes mean that the model suggested above would not be a good estimate of transmembrane water exchange, as it would underestimate the surface-to-volume ratio of the entire CSF space.

– The use of an IVIM signal model is well-motivated by the comparison to vascular density as quantified by histological staining. However, the biological validity of IVIM and bi-exponential signal models in tissue, particularly GM (see recent work on SANDI and other GM signal models for DWI), remains contentious. To help validate the use of IVIM metrics beyond the comparison to standard ADC fitting provided here, an additional analysis of the distribution of apparent diffusivities using an inverse Laplace transform is suggested. This analysis could be carried out only in ROIs where changes in IVIM metrics were observed (namely the olfactory area) to validate (i) a bimodal distribution of diffusivities and (ii) the approximate fractions of slow/fast diffusivities.

We appreciate Reviewer’s remark and we understand the concern regarding efficacy of IVIM model in separating fast microcirculatory from relatively static water component (Wu D and Zhang J 2019, https://pubmed.ncbi.nlm.nih.gov/31267578/). As requested, we performed supplementary voxel-wise inverse Laplace transform (ILT) analysis using a recent ‘INtegrated SPECTral component estimation and mapping’ (INSPECT) algorithm (Slator et al. 2021, https://pubmed.ncbi.nlm.nih.gov/33934005/). Based on unsupervised learning technique, the algorithm provides a possibility of estimating canonical bases for separation of several microstructural environments with not high requirements regarding signal-to-noise ratio of analyzed images.

1) In response to the Reviewer’s remarks, we clarify that we have intentionally analyzed DWI with monoexponential and biexponential fitting approaches to assess mainly the interstitial fluid space in two genotypes. Both approaches possess clinical and translational applicability, and are feasible considering nonisotropic voxel dimensions in acquired EPI images (0.15 × 0.15 × 0.5 mm^3^). In response to the remark, we have provided supplementary analysis using inverse Laplace (ILT) transform method to verify the distribution of diffusivities in depicted DW signals. Nevertheless, this analysis requires further supplementary validation including brain-wide assessment and phantom/synthetic signal analysis before its feasibility, in our suboptimal for this purpose setup including non-segmented EPI in live animals, can be confirmed. Therefore, we kindly request on keeping the analysis only with the response to the remarks.

Summary of the analysis: As suggested, the analysis assuming presence of 1, 2 and 3 spectral components was performed in DW signals derived voxel-wise from 3 regions, where differences in IVIM scores were found: cortical olfactory (OLF), parenchymal caudate (CP), and 3^rd^ ventricular CSF space ROI. Summary of the analysis is provided in the supplementary Figure attached above, and includes assessment for each diffusion-encoding direction separately (‘Diffusion-encoding direction-wise’), all diffusion directions jointly (‘Aggregated distribution’), and for the average signal from all diffusion encoding directions (‘Average distribution’). All spectral components assumptions provided similar estimates of mean ADC so only the results assuming 2 spectral components were used as representative.

We confirm the presence of two main signal clusters mostly visible in ADC spectra from each diffusion-encoding direction separately (Figure above, ‘Diffusion-encoding direction-wise’), from all 3 ROIs: (1) slow and free diffusivity (0.1-3 µm^2^/ms); (2) fast diffusivity (perfusion-like) (>10 µm^2^/ms). Due to slight differences in spectra compartmentation, most probably due to tissue architecture-specific findings in each encoding direction (i.e., neuronal tracts orientation) and/or partial volume from the non-isometric voxel size, the 2^nd^ cluster is visible in the mean spectrum only as a small peak using ‘aggregated’ and is averaged out using the ‘average distribution’ approach. This supports the high correlation we found between average ADC a D measures ROI-wise (Figures 2A and Appendix-figure 1B from the original manuscript).

Using both ‘aggregated’ and ‘average distribution’ approaches, we found two separate components in slow diffusivity ranges in both cortical and parenchymal ROIs. First, mostly of lower magnitude or width in the range of restricted tissue diffusion (<0.6 µm^2^/ms), may reflect several biological restricted compartments. Explanation of its origin would require additional investigation using inversion-recovery technique (which has been presented previously i.e., by Slator et al. 2021a, https://pubmed.ncbi.nlm.nih.gov/33934005/ or Slator et al. 2021b, https://pubmed.ncbi.nlm.nih.gov/34411331/), and validation in the phantom or in synthetic DWI signal. Second, mostly of higher amplitude or width in the lower range of free water diffusion (~1 µm^2^/ms), likely reflects the signal from interstitial fluid space.

Ratio of two spectral components fractions were found not different between Aqp4 (+/+) and Aqp4 (-/-) mice in all 3 regions (see panel D in the Figure above). Additionally, the differences in ADCs between two genotypes (Figure 2A and B from the manuscript) were also associated with higher magnitude or slight shift of spectral components towards the range of free water diffusion in Aqp4 (-/-), reflecting most probably the enlarged ISF space.

However informative, we kindly request the Reviewer for keeping this supplementary analysis only among the responses to the remarks. Our ILT findings require further analysis including a brain-wide investigation, validation in the phantom and synthetic DW signals to provide technical improvements and solid conclusions. This is beyond this current general report, and should be fulfilled by a separate and detailed technical study.

In response to the remark we supplementary discussed our DWI approach and modelling in the manuscript:

Materials and methods, page 15, lines 419 to 429: ‘It is worth mentioning, that there is an ongoing debate on efficacy of IVIM modelling in reflecting phenomena in microvascular network or related to tissue microarchitecture (Fournet et al. 2017, https://pubmed.ncbi.nlm.nih.gov/27903921/; Niendorf et al. 1996, https://pubmed.ncbi.nlm.nih.gov/8946350/; Meeus et al. 2016, https://pubmed.ncbi.nlm.nih.gov/27545824/; Pashoal et al. 2018, https://pubmed.ncbi.nlm.nih.gov/30221622/; Schneider et al. 2019, https://pubmed.ncbi.nlm.nih.gov/31131482/).

[…]

However indicatively useful, higher order models and the models focused on separating hindered MR diffusion signal according to assumption on microarchitecture (Latour et al. 1994, https://www.ncbi.nlm.nih.gov/pmc/articles/PMC43130/; Pfeuffer J et al. 1998, https://pubmed.ncbi.nlm.nih.gov/9608585/; Palombo et al. 2020, https://pubmed.ncbi.nlm.nih.gov/32289460/, Burcaw et al. 2015, https://pubmed.ncbi.nlm.nih.gov/25837598/; Kaden et al. 2016, https://pubmed.ncbi.nlm.nih.gov/27282476/; Wu and Zhang 2019, https://pubmed.ncbi.nlm.nih.gov/31267578/; Olsen et al. 2022, https://pubmed.ncbi.nlm.nih.gov/35168088/) or assessment of diffusion signal distribution (Ronen et al. 2006, https://pubmed.ncbi.nlm.nih.gov/16410179/; Roth et al. 2008, https://pubmed.ncbi.nlm.nih.gov/17574364/; Benjamini and Baser 2019, https://pubmed.ncbi.nlm.nih.gov/30584915/; Slator et al. 2021, https://pubmed.ncbi.nlm.nih.gov/33934005/) are beyond presented general evaluation.’

– Related to the above, IVIM metrics can vary depending on the TE and/or gradient amplitude, which may make comparisons between IVIM studies difficult. See, for instance: 10.1002/mrm.28996. This should be mentioned or addressed.

We agree that these parameters are important for our experiments. Here, we have included higher averaging of b-values ≥1000 s^2^/mm to reduce the possible bias from noise. Additionally, we have applied higher echo time (30 ms, TE). This echo time was chosen based on preliminary in vivo experiments (unpublished) where we found reduced influence of ghosting as well as perfusion-related artefact, and optimal image quality compared to DWI at minimal (22 ms) and long (55 ms) TE.

We have provided additional description in the manuscript:

Methods, page 15, lines 424 to 431: ‘[…] We have aimed to provide an optimal setup for DWI (Lemke et al. 2011, https://pubmed.ncbi.nlm.nih.gov/21549538/; Liao et al. 2021, https://pubmed.ncbi.nlm.nih.gov/33692677/) by measuring MR diffusion signal using 30 ms echo time (TE) and 17 b-values with increased averaging for b-values ≥1000 s^2^/mm (Table 1B). Based on or preliminary assessment (unpublished) application of higher than minimal available (here minimal ~22 ms) TE would reduce the influence of ghosting and perfusion-related artefact, and higher averaging would reduce the influence of possible Rician noise at high b-values. Although IVIM estimates were reported to depend on TE (Bisdas and Klose 2015, https://pubmed.ncbi.nlm.nih.gov/25475914/; Fuhres et al. 2022, https://pubmed.ncbi.nlm.nih.gov/34453445/), mostly for perfusion fraction, our evaluation focused predominantly on the slow diffusion component. Furthermore, by sampling signal up to 3000 s^2^/mm b-value, which may lead to presented slight lower ADC values, we aimed for depicting dominant signal from extracellular space at high b-values (Cihangiroglu et al., 2009, https://pubmed.ncbi.nlm.nih.gov/18162352/; Le Bihan 2019, https://pubmed.ncbi.nlm.nih.gov/29277647/; Niendorf et al. 1996, https://pubmed.ncbi.nlm.nih.gov/8946350/; Clark and Le Bihan 2000, https://pubmed.ncbi.nlm.nih.gov/8946350/). […]’

– On line 289, the authors discuss that WT expression of AQP4 is localized to areas of enlarged interstitial volume and suggest that this may be the result of a compensatory mechanism. References to recent studies of dynamic relocalization of AQP4 and some elaboration concerning the potential signalling pathway would strengthen this suggestion – e.g., see Salman et al. (2022) https://doi.org/10.1093/brain/awab311

We agree that dynamic reorganization of AQP4 may be a primary mechanism of the interstitial volume regulation, and ultimately glymphatic function. We have published evidence for this in vivo, where day/night differences in glymphatic function was associated with increased localization but not expression of AQP4 (Hablitz Nat Commun 2020). We have included this point in our manuscript.

Introduction, page 1, lines 45-47: ‘[…] ‘Thus, the evidence suggests that it is the vascular polarized AQP4 expression in the astrocytic vascular endfeet functionally crucial for fluid transport, and not necessarily total AQP4 levels in the tissue (Sylvain et al. 2021, https://pubmed.ncbi.nlm.nih.gov/33561476/; Amiry-Moghaddam et al. 2003, https://pubmed.ncbi.nlm.nih.gov/12578959/; Mestre et al. 2018, https://www.ncbi.nlm.nih.gov/pubmed/30561329; Hablitz et al. 2020 https://pubmed.ncbi.nlm.nih.gov/32879313/; Eide and Hansson 2018, https://pubmed.ncbi.nlm.nih.gov/28627088/).’

– For the DCE results, the authors argue that the shorter duration of interstitial tracer accumulation in KO vs. WT can be explained by decreased tracer penetration into the brain. Perhaps this could be quantified by looking at the cumulative area under the curve vs. time as a proximal measurement of tracer influx/efflux (although the use of arbitrary intensity units may complicate such an analysis). The data is convincing as presented, but there may be a more succinct visualization/analysis that reveals impaired tracer influx and efflux in the KO mice.

Thank you, we agree that area under the curve (AUC) may be informative, and that the usage of arbitrary intensities from MR contrast agent might, indeed, complicate the cumulative analysis. To overcome this and to address the Reviewer’s request, we have calculated AUC under the DCE-curve for every ROI from each animal, and included these measurements in the Table 3, as well as the manuscript text.

Materials and methods, page 22, lines 717 to 720: […] Duration of significant from baseline interstitial tracer accumulation was obtained as a time difference between the last and the first time points, where DCE signal increase was significantly different from baseline. Area under the DCE curve was calculated for each subject as a sum of DCE signal amplitudes over all the measured time points. […]

Results, page 8, lines 224 to 225: ‘This would be confirmed with the area under the DCE curve that consistently was smaller in KO (Table 3, ‘Mean AUC’).’

Results, page 9, lines 247 to 250: […] Finally, there no correlation was found between the vascular density and AQP4 expression (both Person’s and Spearman’s correlation absolute value <0.52 and p>0.16), or between MR diffusion and tracer dynamics parameters (correlation value <0.5 and p>0.05 for any comparison) except for low correlation between the area under the DCE curve and ADC for both genotypes (WT: r=0.63, p=0.0272; KO: r=0.64, p=0.0239 ). […]

– In some of the ROIs, the DCE time series appears to show two peaks (Figure 3: cisterna magna, Circle of Willis, superior sagittal sinus). Is there any significance or interpretation of the shape of these time series?

Certainly, we hurry to clarify that two peaks were previously observed (Iliff et al. 2013, https://pubmed.ncbi.nlm.nih.gov/23434588/ – Figure 2A; Decker et al. 2022, https://pubmed.ncbi.nlm.nih.gov/34905509/ – Figure 2A; Stanton 2021, https://pubmed.ncbi.nlm.nih.gov/33426699/ – Figure 2I and 3J), though not highlighted. Here, we have intentionally used a higher concentration of gadobutrol (20mM/ml, 1µL/min during 10 minutes) to achieve signal enhancement in the parenchymal brain regions. For high spatiotemporal resolution imaging presented, we employed a steady-state free precession sequence (Scheffler 2004, https://pubmed.ncbi.nlm.nih.gov/15170841/; Bieri and Scheffler 2013, https://pubmed.ncbi.nlm.nih.gov/23633246/) which enables much higher signal-to-noise ratio compared to fast low flip angle gradient echo. This might elucidate likelihood of double peaks appearance in large CSF compartments, close to the infusion site.

– I understand that the CSF segmentation algorithm was developed in-house and will be evaluated and presented in full in a later technical report. At this stage, it may nonetheless be helpful to provide the output of each step in the segmentation process as a supplementary figure – at least for a single ROI.

The Reviewer is correct, we will be publishing our methodology soon. In response, we have supplementary cited the first presentation of the algorithm at ESMRMB 2021 conference (Gomolka et al. 2021, ref. 109: Poster: ‘CSF space volumetry using 3D-CISS in AQP4-deficient mice – quantitative analysis and technical advances’).

In this manuscript, we have provided detailed description of the algorithm, allowing it for replication. The approach computes an initial CSF segmentation map by applying a bounding-box and adaptive thresholding. As thresholding itself is the worst segmentation method, obtained CSF map is used for subsequent redefinition and fine-tunning of the CSF space borders in 3 consecutive steps (each for anatomical orientation, mostly inclusion of missing and removal of obsolete voxels) based on a high contrast to surrounding brain parenchyma. Final segmentation 3D maps (volumes) are presented in the Figure 1 and Appendix-figure 1. In response, we have attached the 3D reconstructions of intermediate steps for a representative 3D-CISS volume in the Appendix-figure 3.

Reference to the figure was added to the Materials and methods section (page 19, lines 568, 582 584, 611, and page 20, lines 632, and 635).

Reviewer #3 (Recommendations for the authors):1. Figure 2 is very dense, making it difficult for the reader to interpret results, authors should consider separating the diffusion-weighted MRI section from the histology section to improve readability.

We are very thankful for the remark. We have reformatted the figure by increasing the spaces to provide visual separation between the panels and increased the font of key labels. The modified figure is attached to the revised manuscript.

2. The ADC values presented in Figure 2C are lower than ADC generally measured in the mouse brain, could the author comment on why they see an underestimation in their work?

Thank you for the remark. Indeed, our ADC values are lower than usually measured in the mouse brain due to combination of measured b-value up to 3000 s2/mm in combination with slightly higher TE. Higher TE was chosen to minimize ghosting as well as influence of arterial pulsations and magnitude of perfusion-related signal, while the high b-value were chosen to focus our analysis mainly on slow MR diffusion measures. We have now addressed this concern in the manuscript.

Methods, page 15, lines 424 to 431: ‘[…] We have aimed to provide an optimal setup for DWI (Lemke et al. 2011, https://pubmed.ncbi.nlm.nih.gov/21549538/; Liao et al. 2021, https://pubmed.ncbi.nlm.nih.gov/33692677/) by measuring MR diffusion signal using 30 ms echo time (TE) and 17 b-values with increased averaging for b-values ≥1000 s^2^/mm (Table 1B). Based on or preliminary assessment (unpublished) application of higher than minimal available (here minimal ~22 ms) TE would reduce the influence of ghosting and perfusion-related artefact, and higher averaging would reduce the influence of possible Rician noise at high b-values. Although IVIM estimates were reported to depend on TE (Bisdas and Klose 2015, https://pubmed.ncbi.nlm.nih.gov/25475914/; Fuhres et al. 2022, https://pubmed.ncbi.nlm.nih.gov/34453445/), mostly for perfusion fraction, our evaluation focused predominantly on the slow diffusion component. Furthermore, by sampling signal up to 3000 s^2^/mm b-value, which may lead to presented slight lower ADC values, we aimed for depicting dominant signal from extracellular space at high b-values (Cihangiroglu et al., 2009, https://pubmed.ncbi.nlm.nih.gov/18162352/; Le Bihan 2019, https://pubmed.ncbi.nlm.nih.gov/29277647/; Niendorf et al. 1996, https://pubmed.ncbi.nlm.nih.gov/8946350/; Clark and Le Bihan 2000, https://pubmed.ncbi.nlm.nih.gov/8946350/). […]’

3. Authors could remove the MRI diffusion phantom work from Figure 2 and include it as an additional Supplementary figure. While it complements the work, it is fairly well established that ADC increase is related to the extravascular space.

Thank you very much for this remark. Indeed, as it is accepted that increasing ADC mainly relates to the increase in interstitial / extravascular space volume of various morphological or functional origin, we did our best to remove the phantom section or to separate the ex vivo microscopy results from MR-DWI results. We found this providing additional separation of DWI-related findings and in confirming stability of our measurements, requiring the potential reader to switch between two figures when focusing on complementary descriptions.

Therefore, to improve the clarity of the figure, we have reformatted the figure by increasing the spaces between and slightly more visual separation between the panels, and by increasing the font of key labels. The modified figure is included in the revised manuscript.

4. Could authors include regional brain volume measurements?

We agree with the Reviewer’s concern. We have mainly focused on the water brain content, and thus the design of our MRI protocol encompassed sequences providing high water signal, but rather low tissue contrast. Hence, CISS and DWI using EPI are not designed to provide accurate measurements of regional brain morphometry due to low parenchymal contrast or associated magnetization and wrapping / ghosting artifacts.

In response to Reviewer’s request, we have attempted a supplementary analysis based on affine registration of Allen Brain atlas template (used for DCE-MRI evaluation) subject-wise to available 3D-CISS images. We found no statistical difference between the regional volumes from *Aqp4 (+/+)* and *Aqp4 (-/-)* mice (Figure below), for all 15 parenchymal regions analyzed in our original manuscript. However, it cannot be excluded that such approach is affected by additional registration-related artefacts and corregistration with common template would assume similar proportionality between the parenchymal subregions in both genotypes. This is a huge caveat, as AQP4 expression is not homogenous within the brain in normal conditions (see Figure 2H). Therefore, we kindly request on keeping the provided morphometry analysis outside the manuscript and only within the responses to the reviewer’s remark.

To provide further clarification of our MRI protocol purpose, we have renamed the 3D-CISS protocol as ‘CSF space volumetry and cisternography’ in every mention in the manuscript, compared to the previous ‘volumetry and cisternography’ (page 21, line62; page 12, line 293; page 15, line 404, 412 and 413; page 19, line 569; page 24, line 807).

5. Page 11, Line 238: "possible reflecting that AQP4 is upregulated in response to stagnation of interstitial fluid" may be too strong a speculation as an increase in ADC is likely suggesting a higher rate of fluid movement. And again on page 13, line 289: "compensatory upregulation of AQP4 to fluid stagnation consistent with the notion that AQP4 reduce parenchymal resistance to water and solute movement" seems contradictory as AQP4 water channel is known as the path of least resistance for water transfer (Manley et al., Nature Medicine, 2000; Papadopoulos et al., Faseb J, 2004), therefore it does not follow that an upregulation would lead to fluid stagnation.

Thank you for this insightful comment. The correlation of increased AQP4 expression and ADC values was meant to emphasize the hypothesis that, in animals that are not congenitally missing this channel, AQP4 prevents fluid stagnation in areas with larger interstitial fluid volume. We have edited the sentence to clarify this point.

Results, Page 9, lines 251-253: ‘[…] Also, the correlation between slow MR diffusion, ADC and D, with AQP4 density across 10 regions in WT mice (Figure 2I) suggests that AQP4 expression is higher in regions with relatively larger interstitial fluid volume, possibly reflecting that AQP4 is upregulated in response to stagnation of interstitial fluid in wildtype mice.’

6. Page 14, Line 312: "Instead, we show that slow diffusion of water across physiological membrane is increased due to more water exchange in an enlarged interstitial space" could be more precise. For example, previous work has shown that blood-brain barrier water exchange is reduced due to the removal of AQP4 (Urshihata et al. Eur Radiol Exp, 2021; Ohene et al., Neuroimage, 2019).

Thank you. We have reformulated the fragment accordingly.

Page 13, lines 333-340 were corrected and expanded with relevant descriptions

‘Instead, we show that slow MR diffusion measures are increased mostly due to an enlarged interstitial space. Only slight 5-6% increase in the mean ADC and D might result from superposition of opposing effects of reduced transmembrane permeability (reducing ADC) and increased extracellular space (increasing ADC) as concluded previously using time-dependent diffusion MRI and Latour’s model of long-time diffusion behavior (Pavlin et al. 2017, https://pubmed.ncbi.nlm.nih.gov/28039592/). Similarly, evaluation of ADC using multi-b-value-multi-diffusion-time DWI provided higher ADC’s in healthy hemispheres of mice subjected to contralateral ischemic stroke, reflecting larger interstitial space in KO (Urushihata et al. 2021, https://pubmed.ncbi.nlm.nih.gov/34617156/). The enlarged interstitial space in both awake and anesthetized AQP4 KO mice is also consistent with previous reports under anesthesia (Yao et al. 2008, https://pubmed.ncbi.nlm.nih.gov/18495879/; Amiry-Moghaddam et al. 2003, https://pubmed.ncbi.nlm.nih.gov/14597704/; Papadopoulos et al. 2004, https://pubmed.ncbi.nlm.nih.gov/15208268/). Our findings in the water phantom filled with Sephadex microbeads of similar porosity but different sizes also confirmed increase in both ADC and D resulting from increased free fluid volume (Figure 2D).’

7. Would the authors be able to provide values for the pseudo-diffusion (D*) measurements?

Yes, we have provided all the calculated diffusion parameters for every subject into the supplementary.xls file including all data acquired. We have additionally referred to this supplementary file in the methods section page 21, line 699: ‘[…] (see calculated values in the supplementary Excel file ‘Figure 1- Source data 1’) […]’.